# JANUSCODER: TOWARDS A FOUNDATIONAL VISUAL-PROGRAMMATIC INTERFACE FOR CODE INTELLIGENCE

**Qiushi Sun**[♡◇†*] **Jingyang Gong**[♡*] **Yang Liu**[☾*] **Qiaosheng Chen**[☾*] **Lei Li**[✫] **Kai Chen**[◇]
**Qipeng Guo**[◇⌀✉] **Ben Kao**[♡] **Fei Yuan**[◇]
[♡]The University of Hong Kong [◇]Shanghai AI Laboratory [☾]Nanjing University
[✫]Carnegie Mellon University [⌀]Shanghai Innovation Institute
qiushisun@connect.hku.hk, jingyang.gong@nyu.edu
qschen@smail.nju.edu.cn, yliu20.nju@gmail.com, leili@cs.cmu.edu
kao@cs.hku.hk, {chenkai,guoqipeng,yuanfei}@pjlab.org.cn

## ABSTRACT

The scope of neural code intelligence is rapidly expanding beyond text-based source code to encompass the rich visual outputs that programs generate. This visual dimension is critical for advanced applications like flexible content generation and precise, program-driven editing of visualizations. However, progress has been impeded by the scarcity of high-quality multimodal code data, a bottleneck stemming from challenges in synthesis and quality assessment. To address these challenges, we make contributions from both a data and modeling perspective. We first introduce a complete synthesis toolkit that leverages reciprocal synergies between data modalities to efficiently produce a large-scale, high-quality corpus spanning from standard charts to complex interactive web UIs and code-driven animations. Leveraging this toolkit, we construct JANUSCODE-800K, the largest multimodal code corpus to date. This powers the training of our models, JANUS-CODER and JANUSCODERV, which establish a visual-programmatic interface for generating code from textual instructions, visual inputs, or a combination of both. Our unified model is a departure from existing approaches that build specialized models for isolated tasks. Extensive experiments on both text-centric and vision-centric coding tasks demonstrate the superior performance of the JANUSCODER series, with our 7B to 14B scale models approaching or even exceeding the performance of commercial models. Furthermore, extensive analysis provides key insights into harmonizing programmatic logic with its visual expression.

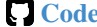 **Code**          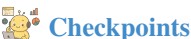 **Checkpoints**          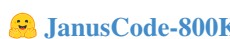 **JanusCode-800K**

## 1 INTRODUCTION

The advent of Large Language Models (LLMs; Hurst et al., 2024; Anthropic AI, 2024) has significantly advanced the field of code intelligence (Sun et al., 2024a), revolutionizing tasks centered on textual source code. Building on this, the scope of code intelligence naturally expands beyond text to encompass the rich and diverse visual manifestations that programs generate (Gemini Team, 2025; Si et al., 2025), with the aspiration of bridging the perceptual–symbolic gap. Establishing a generalist modeling interface that harmonizes code's logic with its visual expression is therefore the next frontier. Such an interface would empower models to flexibly generate data visualizations (Galimzyanov et al., 2025; Ni et al., 2025) and interactive front-ends (Chen et al., 2025a;b), replicate or precisely edit visual artifacts from multimodal inputs (Yang et al., 2025b; Xia et al., 2025), and even build complex, code-driven animations (Ku et al., 2025) to elucidate a concept like "Attention Is All You Need".

Despite its promise, the connection between code and vision remains in its early stages. While recent models have shown success in handling unimodal symbolic representations (Xu et al., 2024b), extending this to multimodal scenarios presents far greater challenges. The first challenge lies at the modeling level. Current research predominantly focuses on program-aided understanding (Qiu et al.,

---

[*]Equal contribution [✉] Corresponding author [†]Project lead

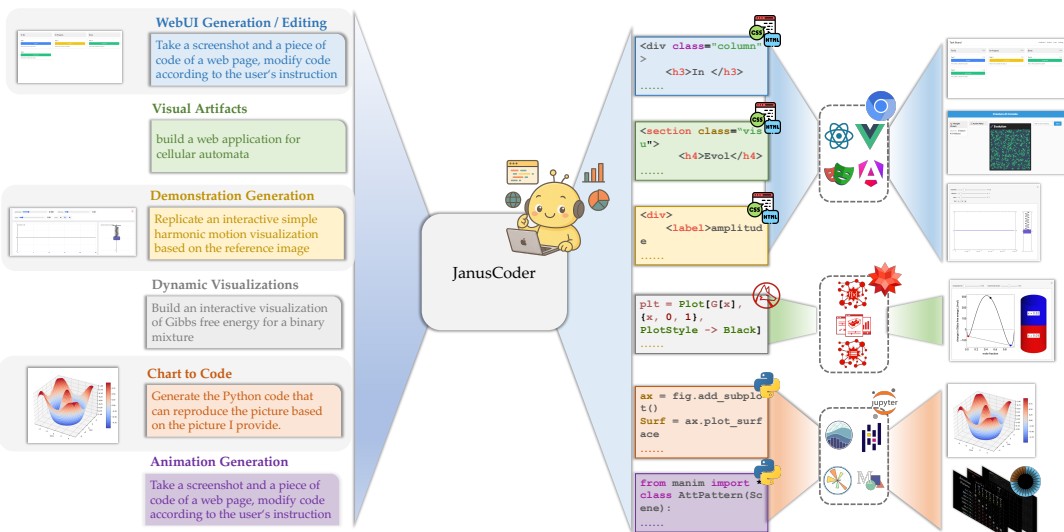

Figure 1: JANUSCODER is a suite of models that establishes a unified visual-programmatic interface, advancing multimodal code intelligence. It supports diverse tasks by combining code with visual content generation, editing, and interpretation in a unified manner.

2025; Chen et al., 2025c) and reasoning (Surís et al., 2023; Guo et al., 2025), while fine-grained perception (Liu et al., 2025) and generative capability remain significantly underdeveloped (Wang et al., 2025a). For the few well-explored scenarios (Wang et al., 2024; Yun et al., 2024), existing works often build specialized models for isolated targets (*e.g.*, one for chart-to-code, another for WebUI-to-code), leading to models that can neither generalize across scenarios nor scale effectively.

Second, and more fundamentally, progress is impeded by the scarcity of high-quality, diverse multimodal code data. The heterogeneity of content in existing corpora (Gui et al., 2025; Ni et al., 2025) presents a significant challenge, along with varying data richness across different programming languages (PLs), diverse styles of natural language (NL) instructions, and the vast array of visual outputs that code can produce. For instance, these visual outputs can range from static Matplotlib charts and interactive WebUIs to extended animations in the style of 3Blue1Brown[1]. Creating a comprehensive corpus that covers this spectrum is a formidable task. It requires not only large-scale data collection and processing but also well-matched validation environments (*e.g.*, computation / rendering engines), and rigorous quality control over the diverse visual contents.

In this work, we are motivated to build a unified model to facilitate the development of multimodal code intelligence. Toward this goal, we make the following contributions:

1. We develop and release a versatile data synthesis toolkit. This enables the automatic synthesis of multimodal code data across heterogeneous domains and PLs, including but not limited to charts, Web UIs, visual artifacts, and code-driven animations. By doing so, it significantly reduces the engineering efforts required for data curation in future research.

2. Building on this data toolkit, we curate JANUSCODE-800K, the largest multimodal code intelligence corpus to date. Notably, our corpus includes large-scale animation and artifact data that have not been present in previous works.

3. With the above data innovations and by fostering synergies across different modalities and tasks, we developed JANUSCODER and JANUSCODERV. As illustrated in Figure 1, these models constitute a unified interface designed to tackle a broad spectrum of visual–programmatic tasks.

4. We present a comprehensive evaluation, covering seven established and newly proposed benchmarks. Our models demonstrate superior performance improvements in both text-centric and vision-centric settings, approaching or even exceeding the performance of leading commercial models. This indicates that the JANUSCODER series can serve as a strong open-source foundational model for future research and applications.

---

[1] https://www.3blue1brown.com/

## 2 RELATED WORKS

**Code Generation for Visual Interfaces.** LLMs have been widely explored for text-centric code generation of visual interfaces, including data visualizations (Yang et al., 2024; Zhuo et al., 2025b), web pages (Chen et al., 2025d), and interactive UIs (Chen et al., 2025a). Early efforts focused on Python libraries (*e.g.*, Matplotlib, Seaborn) for producing figures in scientific workflows (Zhang et al., 2024b; Sun et al., 2025b). Later work extended to chart generation and editing (Zhao et al., 2025a), and to mapping NL instructions into web-based artifacts (Zhang et al., 2025) or structured UI interactions (Cheng et al., 2024; Sun et al., 2024b). Overall, these approaches highlight the potential of LLMs to author executable visual content, though they remain constrained to text-driven inputs.

**Visually-Grounded Code Generation and Understanding.** Another line of work emphasizes multimodal inputs (vision-centric), where models interpret visual information to produce or reason about symbolic code (Hu et al., 2024; Jiang et al., 2025). Representative efforts include chart understanding, which evaluates the extraction of structured knowledge from plots (Masry et al., 2022; Zhang et al., 2024a), and chart-to-code generation, which requires reproducing scientific plots from images with captions or instructions (Zhao et al., 2025b; Xia et al., 2025; Wu et al., 2025). Beyond charts, studies extend to theorem visualization (Ku et al., 2025), multimodal algorithmic problem solving (Li et al., 2024), and structured vector graphics such as SVGs (Yang et al., 2025c; Nishina & Matsui, 2024). While these works demonstrate progress, they largely target isolated domains and modalities. In contrast, we move beyond these constraints by unifying diverse domains and modalities across charts, web UIs, animations, symbolic computation, and more, taking a leap forward in advancing multimodal code intelligence.

## 3 METHOD

To empower models for multimodal code intelligence, we propose a versatile data toolkit that incorporates model interactions (Sun et al., 2024c) and compiler feedback to tackle multifaceted demands. In contrast to prior data approaches, which often suffer from a lack of instruction diversity, scarcity in specialized domains, and insufficient validation for visual-code alignment, our pipeline establishes a principled workflow. As shown in Figure 2: (1) Data Sourcing, where raw assets are collected and categorized; (2) Data Synthesis & Curation, where new instruction-code pairs is generated and refined through a multi-strategy engine; and (3) Quality Control, which ensures data fidelity through automated validation and LLM/VLM judging.

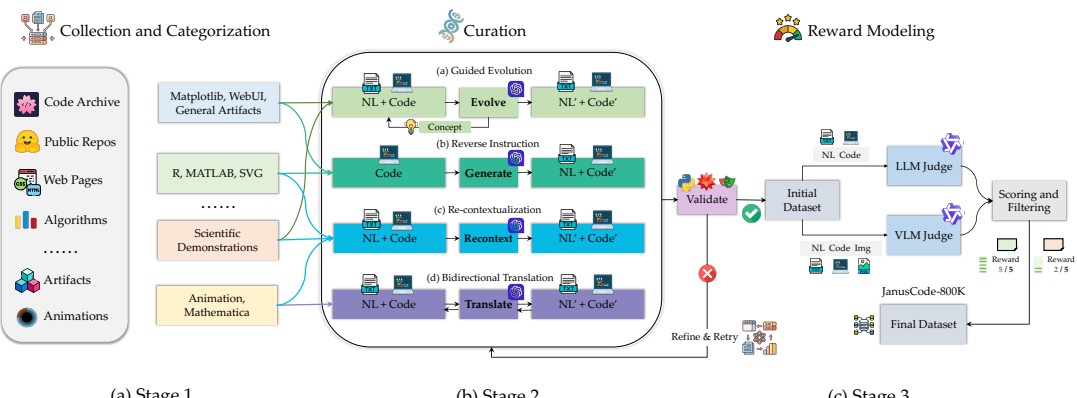

Figure 2: An overview of our toolkit for curating JANUSCODE-800K, which integrates heterogeneous data sourcing, multi-strategy synthesis and curation, and LLM/VLM-based reward modeling after execution checks.

### 3.1 DATA COLLECTION

Our pipeline begins by aggregating raw data from a vast and heterogeneous sources. These include large-scale public datasets (*e.g.*, StackV2; Lozhkov et al., 2024), extensive web corpora (*e.g.*, Web-

Table 1: Overview of the strategies used to construct JanusCode Data from multiple sources, different colored squares represent different strategies: ■ Guided Evolution; ■ Re-contextualization; ■ Reverse Instruction; ■ Bidirectional Translation.

| Source Data Type | Size | Validation | Reward | Strategies |
|---|---|---|---|---|
| Matplotlib | 200K | Python | VLM | ■ ■ |
| Charts | 77K | Python | VLM | ■ ■ |
| Algorithm | 100K | Python | VLM | ■ |
| Mathematica | 11K | Wolfram Engine | LLM | ■ ■ ■ |
| Animation | 5K | Python + Manim Engine | VLM | ■ ■ |
| Scientific PLs | 400K | - | LLM | ■ ■ ■ |
| SVG | 400K | - | VLM | ■ |
| WebUI | 270K | Playwright | VLM | ■ |
| General Artifacts | 10K | Playwright | VLM | ■ ■ |
| Scientific demonstration | 10K | Playwright | VLM | ■ |

Code2M; Gui et al., 2025), specialized knowledge bases like the Wolfram Demonstrations Project, and competitive programming problems (Xu et al., 2025; Sun et al., 2025a). All sourced data is then classified into two primary formats:

- Paired Data ($D_{\text{paired}}$): Datasets containing instruction-code pairs ($I, C$). When a visual output is available, it is included as an optional component, forming a triplet ($I, C, V$).
- Code-Only Data ($D_{\text{code}}$): Unlabeled datasets consisting solely of code snippets, denoted as $C$.

A significant challenge within $D_{\text{code}}$ is the long-form, complex code files, such as a single Manim script that generates a 5-minute-long mathematical animation. Such monolithic files contain numerous distinct conceptual steps but are not structured for direct learning. To address this, we employ a sophisticated decomposition strategy utilizing Abstract Syntax Trees (AST). We parse complex source code into its AST representation and traverse the tree to identify and isolate semantically coherent, self-contained logical units. The details of the preprocessing pipeline and data sources can be found in Appendix A and Appendix B, respectively.

## 3.2 DATA CURATION

We aim to build two complementary types of data: text-centric instruction-code pairs ($I, C$) for tasks like Python data visualization, and vision-centric triplets ($I, C, V$) for tasks such as chart-to-code.

**Guided Evolution.** We adapt our previously proposed interaction-driven synthesis (Sun et al., 2025a) to this strategy, aiming to increase data complexity and diversity. Starting with a seed triplet ($I, C$) $\in D_{\text{paired}}$, the evolution is guided by a high-level concept $\mathcal{K}$, represented as keywords (*e.g.*, chart type) or a web meta-task (*e.g.*, 'add a widget'). A new instruction is generated via $I' = f_{\text{evolve}}(I, C, \mathcal{K})$. This conceptual guidance is critical for creating grounded and novel instructions for visual coding tasks that move beyond simple heuristic-based evolution (Xu et al., 2024a). Subsequently, the model generates code $C'$ for the new instruction, which is then validated in an execution environment $E$. The feedback from this validation step drives the next synthesis iteration.

**Re-Contextualization.** This method enhances the semantic quality of existing paired data, maximizing the utility of our verified code assets. For a given pair ($I, C$) $\in D_{\text{paired}}$, $f_{\text{recontext}}$ performs a deep analysis of the code $C$ to uncover implicit logic, edge cases, or contextual details not specified in the original instruction $I$. It then generates a more descriptive and precise instruction, $I' = f_{\text{recontext}}(I, C)$. The primary strength of this approach is its efficiency; it creates a higher-fidelity pair ($I', C$) by improving the quality of the instruction without the computational overhead of synthesizing and validating entirely new code. This ensures the model is trained on a semantically richer dataset where language and code are more tightly aligned.

**Reverse Instruction.** The primary value of this strategy lies in its ability to transform raw code into aligned instruction-code pairs, thereby substantially expanding data coverage. Inspired by prior practices that exploit large-scale open-source code to synthesize realistic tasks (Wei et al., 2024), we develop a reverse-instruction process: given a reference file $C_{\text{ref}} \in D_{\text{coder}}$ a snippet of $K$ lines $C_{\text{sample}}$ is sampled and passed to a function $f_{\text{reverse}}$ to produce a plausible natural language instruction $I'' = f_{\text{reverse}}(C_{\text{sample}})$. A model then generates $C'$ conditioned on $I'$, optionally leveraging $C_{\text{ref}}$ as broader context. This pipeline enables the systematic repurposing of theorems and data analysis

code from scientific PLs like R and Matlab into instruction-following samples ( $I'$, $C'$ ), effectively populating our dataset with a rich variety of domain-specific tasks.

**Bidirectional Translation.** This strategy fosters the learning of abstract, syntax-independent representations by translating conceptual intent between semantically analogous domains (e.g., Manim and Mathematica), effectively multiplying the value of our specialized datasets. Given a sample ( $I^A$, $C^A$ ) from a source domain A , a new instruction for the target domain B is first generated: $I^B = f_{\text{translate}}\left(I^A\right)$. Subsequently, the model generates the target code $C^B$ that uses the source code $C^A$ as a structural template: $C^B = f_{\text{translate}}\left(I^B, C^A\right)$. This approach pragmatically addresses the challenge of generating complex code from scratch. The process is fully bidirectional.

After data curation, next component of our toolkit is the validation of synthesized code. We leverage a sandbox $E$ that provides the necessary backends (*e.g.*, Python interpreters, web renderers). Every newly generated code sample $C'$ must pass through a formal execution function, $V' = \text{Exec}\left(C', E\right)$, to produce a visual output or pass collected / generated test cases. This step ensures that only functionally correct code proceeds to the final quality control stage. Samples that fail this validation are rerouted to the synthesis engine for retry and refinement.

### 3.3 CROSS-DOMAIN SYNERGIES

Rather than treating data sources in isolation, we deliberately exploit synergies across heterogeneous domains and modalities. The central idea is that knowledge can be transferred between semantically related domains (*e.g.*, R code reinforcing Mathematica tasks) and across different modalities (*e.g.*, the visual output of a Python data visualization task can be used to construct chart-to-code data). This approach is highly effective for mitigating data scarcity in specialized areas, such as scientific demonstration, and enhances the overall coverage and robustness of our dataset.

This principle is applied throughout our data curation process. For instance, the wealth of scientific computing logic in R and Matlab corpora is generalized to synthesize new data for Manim and Mathematica using our Reverse Instruction and Bidirectional Translation strategies. Similarly, foundational data from WebDev, including HTML and SVG code, provides a robust basis for generating complex, interactive scientific demonstrations. This synergy is crucial for broadening task diversity and strengthening model generalization, as we discuss further in Section 6.1.

### 3.4 DATA QUALITY CONTROL

While our synthesis pipeline generates substantial executable text-centric and vision-centric code, executability alone is an insufficient proxy for the quality of the generated visual content. It is crucial to recognize that while a program may pass compiler or rendering checks, its actual visual output can drastically diverge from user instructions or requirements. We therefore construct a reward modeling pipeline, tailored to our different data types, to systematically assess and filter out misaligned or low-quality data at scale.

Our reward model employs a VLM as its core engine to assess the quality of data. The reward process, denoted by the function $R$, takes NL instruction $I$, the generated code $C$, and the resulting visual output $V$. These elements are organized within a structured prompt that guides the VLM through a two-stage evaluation: (1) task understanding, where it summarizes its interpretation of the instruction, and (2) Multi-dimensional Rating & Scoring across the four key metrics of task relevance, task completion, code quality, and visual clarity.

Each metric is assigned an integer score on a scale of [1-5]. The final reward score $S$ is calculated as the average of these scores: $S = R(I, C, V)$. Only data samples whose score $S$ exceeds a predefined threshold are retained. For data without a visual output $V$, a similar process is employed using an LLM to assess the ( $I$, $C$ ) pair.

### 3.5 JANUSCODE-800K

Leveraging our data toolkit, we construct JANUSCODE-800K, a diverse and high-quality multimodal code intelligence corpus that we will release to the community. To the best of our knowledge, it is the largest and most comprehensive of its kind to date. The detailed statistics are presented in Table 2.

Table 2: Statistics of JANUSCODE-800K.

| Data Type | Statistics |
|---|---|
| **Text-centric** | |
| Python Visualization: Generation | 127.5K |
| Python Visualization: Editing | 51.8K |
| Scientific PLs | 31.8K |
| SVG | 20.0K |
| Animation | 19.5K |
| General Artifacts | 56.8K |
| Algorithm Data | 100.0K |
| **Vision-centric** | |
| Chart-to-Code | 70.0K |
| WebUI Generation | 200.0K |
| WebUI Editing | 69.5K |
| Scientific Demonstration | 53.0K |

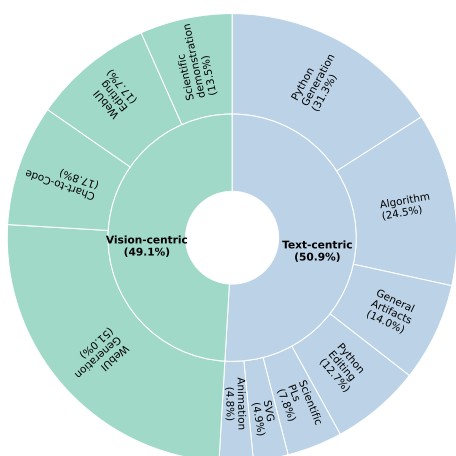

Figure 3: Distribution of JANUSCODE-800K.

In terms of its composition, we achieve a balance between the amount of text-centric and vision-centric data. The overall distribution of task types is shown in Figure 3. During training, JANUSCODERV utilizes the entire corpus, while JANUSCODER is trained exclusively on the text-centric data.

## 4 DTVBENCH

We present DTVBENCH for evaluating the capability of models to generate code for *dynamic* theorem visualizations. The benchmark integrates two complementary engines: (i) MANIM, an engine for creating explanatory mathematical animations, and (ii) WOLFRAM MATHEMATICA (Wolfram Research, 2025), a symbolic computation engine supporting interactive visualizations. By combining these two domains, DTVBENCH assesses a model's ability to translate NL instructions into dynamic, logically coherent, and visually faithful visualizations of theorems.

### 4.1 DATA COLLECTION AND CURATION

We obtain raw samples from human-authored and verified sources and preprocess them following the method in Section 3.1. Tasks in DTVBENCH are derived from code such as 3BLUE1BROWN video segments and official Wolfram demonstrations. From these sources, we manually curated 102 visualization tasks for the benchmark.

### 4.2 EVALUATION

We adopt a multi-dimensional evaluation protocol for both engines. Each generated output is scored along the following dimensions:

- **Executability** ($s_{exec} \in \{0, 1\}$): whether the generated code can be successfully executed.
- **Code Similarity** ($s_{sim} \in [1, 5]$): structural and syntactic consistency with the reference solution, judged by GPT-4o.
- **Instruction Alignment** ($s_{align} \in [1, 5]$): semantic consistency between the natural language instruction and the produced output, judged by GPT-4o.
- **Faithfulness** ($s_{faith} \in [1, 5]$): since dynamic content is primarily intended for human interpretation and interactive outputs are difficult for LLM-based judges to evaluate, we introduce an optional subjective score assessing the plausibility and visual correctness of the generated animation or interactive content.

The overall score is defined as $= s_{exec} \cdot \left(s_{sim} + s_{align} + s_{faith}\right)$. This ensures that only executable code is considered for further evaluation, while successful generations are rewarded for syntactic fidelity, semantic alignment, and perceptual faithfulness. More details of DTVBENCH are in Appendix D.

## 5 EXPERIMENTS

### 5.1 EXPERIMENTAL SETTINGS

**Data Curation and Synthesis.** As described in Section 3, we construct a complete data toolkit to synthesize training data for multimodal code intelligence. All natural language instructions and code are generated using gpt-oss-120b (OpenAI, 2025). For quality control, we adopt reward models with different backbones: Qwen2.5-VL-72B-Instruct (Bai et al., 2025) to evaluate vision-centric data such as Python visualizations and rendered webpages, and Qwen3-235B-A22B (Yang et al., 2025a) to handle text-centric data (*e.g.*, Mathematica code).

**Backbone Models.** For model construction, we use Qwen3-{8B,14B} (Yang et al., 2025a) as the backbones of JANUSCODER, and Qwen2.5-VL-7B-Instruct (Bai et al., 2025) together with InternVL3.5-8B (Wang et al., 2025b) as the backbones of JANUSCODERV. In the analysis part, we additionally include Qwen3-4B, Qwen2.5-Coder-7B-Instruct (Hui et al., 2024), and InternVL3.5-4B for further comparison. Model details are provided in Appendix E.

**Baselines.** Beyond the backbones used by the JANUSCODER series, we include additional baselines for comparison. For unimodal settings, we consider Qwen2.5-Coder-14B-Instruct and Llama-3-8B (Dubey et al., 2024); for multimodal settings, we adopt MiniCPM-V-2-6 (Yao et al., 2024) and Llama-3.2-11B-Vision-Instruct (Meta, 2024). We also report GPT-4o (Hurst et al., 2024) results.

### 5.2 BENCHMARKING

We thoroughly evaluate the JANUSCODER series by employing a broad range of benchmarks that span both unimodal and multimodal code intelligence tasks:

**Unimodal Settings.** Unimodal benchmarks mainly focus on text-to-code generation, including PandasPlotBench (Galimzyanov et al., 2025) for Python visualizations, ArtifactsBench (Zhang et al., 2025) for interactive visual artifacts, and DTVBENCH for dynamic visualization.

**Multimodal Settings.** Multimodal benchmarks cover ChartMimic (Yang et al., 2025b) for chart-to-code tasks, WebCode2M (Gui et al., 2025) and DesignBench (Xiao et al., 2025) for WebUI generation and editing, and InteractScience (Chen et al., 2025b) for scientific demonstration code generation.

**General Coding.** We also evaluate on BigCodeBench (Zhuo et al., 2025a) and LiveCodeBench (Jain et al., 2025) to highlight its capability in following complex instructions and algorithmic capability.

### 5.3 MAIN RESULTS: UNIMODAL TASKS

We first present the results on unimodal tasks in Table 3, where the inputs are mainly NL instructions, code snippets, or both. The outputs are code, which are then executed to generate figures, animations, or rendered webpages for evaluation.

Table 3: Results on PandasPlotBench, ArtifactsBench, and DTVBENCH.

| Model | PandasPlotBench | | | ArtifactsBench | DTVBENCH | |
| --- | --- | --- | --- | --- | --- | --- |
| | Incorrect Code↓ (%) | Visual | Task | | Manim | Wolfram |
| *Open-Source* | | | | | | |
| LLaMA3-8B-Instruct | 26.9 | 59 | 69 | 36.5 | 4.92 | 3.15 |
| Qwen3-8B | 20.0 | 63 | 74 | 36.5 | 6.20 | 5.18 |
| Qwen2.5-Coder-7B-Ins | 21.1 | 63 | 76 | 26.0 | 8.56 | 4.04 |
| Qwen3-14B | 12.6 | 65 | 78 | 39.8 | 6.63 | 5.08 |
| Qwen2.5-Coder-32B-Ins | 12.0 | 66 | 82 | 35.5 | 9.61 | 4.98 |
| JANUSCODER-8B | 14.9 | 63 | 80 | 39.6 | **9.70** | **6.07** |
| JANUSCODER-14B | **9.7** | **67** | **86** | **41.1** | 8.41 | 5.97 |
| *Proprietary* | | | | | | |
| GPT-4o | 9.7 | 72 | 85 | 37.9 | 10.60 | 4.92 |

**Python Visualizations.** We begin by evaluating Python-based visualization tasks (Galimzyanov et al., 2025) where the model generates plotting code from NL descriptions based on DataFrames.

Both our 8B and 14B models show strong performance, exceeding baselines with error rates $< 10\%$, and achieving comparable or superior results to GPT-4o in task completion and visual similarity. Moreover, as unified models, JANUSCODERV also excels in unimodal tasks, as reported in Table 7.

**Visual Artifacts.** JANUSCODER delivers results on ArtifactsBench (Zhang et al., 2025) that are significantly better than GPT-4o, which can be attributed to our data pipeline that combines challenging webdev data for complex interactive components with theorem-related resources and cross-language code to enrich structural diversity and enhance generalization.

**Animations and Interactive Contents.** On DTVBENCH, JANUSCODER also performs strongly in generating dynamic contents, achieving higher code quality and better subjective evaluations than other baselines, approaching the performance of GPT-4o.

## 5.4 MAIN RESULTS: MULTIMODAL TASKS

We then report the results on multimodal tasks in Table 4, where the inputs consist of NL instructions, code, images, or their combinations. The outputs are code, which are subsequently executed or rendered into visualizations or interactive pages for evaluation.

**Chart-to-Code Tasks.** We evaluate JANUSCODERV on ChartMimic (Yang et al., 2025b), JANUS-CODERV achieves strong results on both high- and low-level metrics, consistently outperforming baselines and substantially surpassing GPT-4o. As a unified model, it also outperforms recently released specialized chart-to-code MLLMs (Xia et al., 2025; Zhao et al., 2025b), highlighting the effectiveness of leveraging cross-task data synergy. Detailed comparisons are provided in Appendix G.

Table 4: Results on ChartMimic, DesignBench, WebCode2M, and InteractScience.

| Model | ChartMimic | | | | DesignBench | | WebCode2M | | InteractScience | | |
| | Customized | | Direct | | Gen. | Edit. | Visual | TreeBLEU | Func. | Visual | |
| | Low | High | Low | High | | | | | Overall | CLIP | VLM |
|---|---|---|---|---|---|---|---|---|---|---|---|
| *Open-Source* | | | | | | | | | | | |
| Qwen2.5-VL-7B-Ins | 51.07 | 58.69 | 40.73 | 41.70 | 72.73 | 6.85 | 73.42 | 12.83 | 8.40% | 45.86 | 19.83 |
| InternVL3-8B | 51.88 | 60.04 | 48.48 | 55.41 | 69.34 | 7.76 | **79.62** | 12.40 | 8.93% | 53.35 | 22.05 |
| InternVL3.5-8B | 51.56 | 59.55 | 46.02 | 53.39 | 71.73 | 8.63 | 79.09 | 11.95 | 11.47% | 56.79 | 24.17 |
| MiniCPM-V-2-6 | 27.53 | 48.18 | 21.82 | 45.26 | 66.25 | 4.56 | 45.85 | 9.73 | 0.13% | 20.65 | 7.70 |
| Llama-3.2-11B-Vision-Ins | 18.87 | 39.63 | 19.32 | 28.37 | 62.24 | 6.61 | 51.54 | 6.57 | 6.67% | 32.87 | 13.24 |
| JANUSCODERV-7B | 64.72 | 72.77 | 65.73 | 72.73 | **73.31** | **8.79** | 75.78 | **26.21** | **17.73%** | 60.56 | 27.67 |
| JANUSCODERV-8B | **66.68** | **74.20** | **65.79** | **73.18** | 68.86 | 8.63 | 66.34 | 18.28 | 17.60% | **61.52** | **33.32** |
| *Proprietary* | | | | | | | | | | | |
| GPT-4o | 59.4 | 67.42 | 57.16 | 64.62 | 76.83 | 9.23 | 82.67 | 13.00 | 27.20% | 70.14 | 46.01 |

**Webpage Generation and Editing.** Models are evaluated on generating or editing HTML code to produce webpages grounded in screenshots. In both WebCode2M (Gui et al., 2025) and Design-Bench (Xiao et al., 2025), our models demonstrate significant improvements in both visual quality and the structural similarity of the generated code to the references.

**Scientific Demonstration Generation.** Finally, we evaluate the most challenging and novel task of scientific demonstration code generation (Chen et al., 2025b), which requires the integration of visual understanding, algorithmic reasoning, and spatial comprehension, together with domain knowledge and front-end coding capabilities.

Due to space limitations, the detailed metrics for the results on all the aforementioned benchmarks are presented in Appendix F.

## 6 ANALYSIS

### 6.1 ABLATION STUDIES

**Data Synergies.**    To validate the cross-domain and cross-modal synergies proposed in Section 3.3, we conduct ablation studies by selectively removing specific categories of data within JANUSCODE-800K. The results support our claim, showing that data from non-target domains, even when cross-modal, can provide transferable coding capabilities (Sun et al., 2023) on specialized visual tasks (*e.g.*, text-centric data contributing to multimodal coding scenarios). This provides useful guidance for the research community, suggesting that performance in data-scarce scenarios such as animations and artifacts can be improved by incorporating data from related, more abundant sources.

Table 5: Ablation studies of JANUSCODER and JANUSCODERV across multiple benchmarks. Results marked with [*] indicate evaluations conducted on a subset of the benchmark.

| Method | PandasPlotBench | | ArtifactsBench[*] | LcbV6 |
| | Visual | Task | | |
| --- | --- | --- | --- | --- |
| JANUSCODER | 63 | 80 | 40.99 | 25.14 |
| *w/o Algorithm* | 62↓ | 83↑ | 40.31↓ | 17.71↓↓ |
| *w/o SVG* | 63 | 82↑ | 40.27↓ | 22.86↓ |
| *w/o Rewarding* | 60↓ | 77↓ | 38.58↓ | 24.57↓↓ |

| Method | ChartMimic | InteractScience | WebCode2M |
| --- | --- | --- | --- |
| JANUSCODERV | 68.74 | 17.73 | 75.78 |
| *w/o Algorithm* | 70.16↑ | 18.13↑ | 72.18↓↓ |
| *w/o Chart2Code* | 56.50↓↓ | 16.27↓ | 71.92↓↓ |
| *w/o Text-centric* | 60.73↓ | 12.93↓↓ | 71.82↓↓ |
| *w/o Rewarding* | 58.26↓↓ | 17.20↓ | 73.78↓ |

**Reward Modeling.**    As shown above, we randomly sample from the synthetic data that passes validation but is not filtered by reward modeling. With consistent training set size, we observe a clear performance drop.

This result validates the critical role of our reward modeling for multimodal parts, demonstrating that successful execution alone is insufficient to guarantee high-quality data.

### 6.2 EFFECT OF BACKBONES

To further validate the effectiveness of our data construction, beyond the original experimental setup we additionally adopt Qwen2.5-Coder-7B-Ins and InternVL3.5-4B as backbones. As shown in Figure 4, JANUSCODE-800K consistently yields significant improvements across models with different scales and post-training strategies.

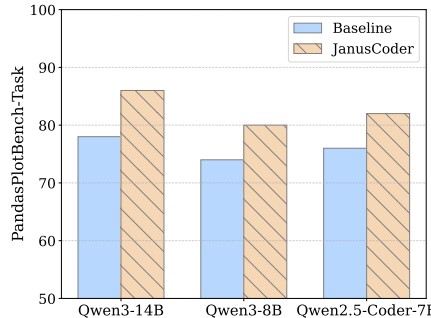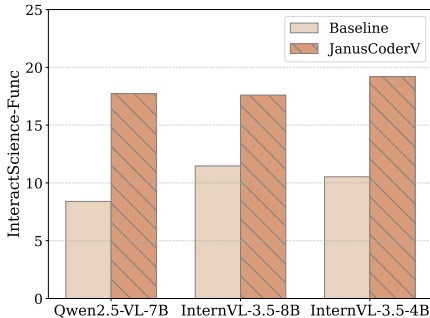

Figure 4: Effectiveness on different model backbones

This confirms the soundness of our data design and can empower diverse backbones to become more generalist models for multimodal code intelligence. More experiments on different backbones are available in Appendix G.2.

### 6.3 GENERAL CODING CAPABILITIES

JANUSCODER demonstrates superior general coding capabilities that surpass even specialist approaches. As shown in Figure 5, it achieves strong performance on general benchmarks while also outperforming specialist models like VisCoder (Ni et al., 2025) in their own target visualization

domain. Furthermore, it outperforms `GPT-4o` in both scenarios, which further demonstrates our model's balanced capabilities. More comparisons are provided in Appendix G.1.

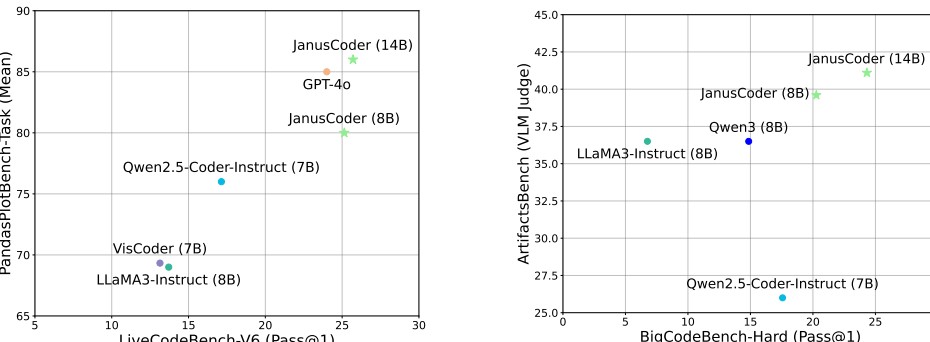

Figure 5: Visualization of balanced visual content generation and general coding ability.

## 7 CONCLUSION

In this work, we introduce JANUSCODER, a suite of foundational models designed to establish a unified visual-programmatic interface. Supported by a complete and scalable data synthesis toolkit, our models handle a diverse spectrum of visual code tasks in a unified manner Extensive experiments on representative benchmarks, including a new benchmark proposed in this work, demonstrate the stunning performance of the JANUSCODER series, with our 7B to 14B scale models approaching or even exceeding the capabilities of leading commercial models. Further analysis reveals the key principles for building such models. JANUSCODER serves as a strong standard for multimodal code intelligence, setting the stage for future advancements in this field.

## ACKNOWLEDGEMENT

We thank Fangzhi Xu and Zhangyue Yin for their valuable feedback on improving the manuscript's presentation. We also express our gratitude to Zichen Ding for the assistance in addressing challenges encountered during the fine-tuning of InternVL 3.5. This research is supported by Shanghai Artificial Intelligence Laboratory and WYNG Foundation (HKU 25AG100407), with sincere appreciation for their support.

## AUTHOR CONTRIBUTIONS

The authors contributed to this work in the following ways: Qiushi Sun led the project, proposing the first feasible version of building a visual-programmatic interface and planning the overall direction. He implemented the initial data pipelines (Python, Matlab, R, Manim, and other PLs/libraries), proposed the data mixture strategy, wrote the main paper, and created the figures. Jingying Gong was responsible for overall model training (unimodal and multimodal) and evaluation of Python-related tasks, and resolved multiple engineering challenges in large-scale code data rollout. Yang Liu implemented the data pipeline and evaluation for HTML-related data, participated in multimodal training, and assisted in formulating the data mixture strategy. Qiaosheng Chen curated the seed data for interactive content and was responsible for the data pipeline and evaluation of demo-related data. Qiushi Sun, Jingyang Gong, Yang Liu, and Qiaosheng Chen all participated in paper reviewing, refinement, and providing demo cases. Lei Li assisted in discussions on code data synthesis. Kai Chen reviewed the submitted manuscript, provided detailed revision suggestions, and discussed strategies for better data utilization. Qipeng Guo oversaw the design of the main data strategy and provided substantial assistance in data pipelines, evaluation, and base model selection, and also offered valuable feedback during the paper rebuttal phase. Ben Kao offered advice on paper writing and impact maximization. Fei Yuan participated in discussions.

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

## A   DATA TOOLKIT DETAILS

### A.1   AST PASRSING

We take the follow steps to process large and complex Manim animations collected from GitHub.

**AST-based Static Analysis.**   We employ a static analysis approach to process the Manim source files without executing them. Each .py script is parsed into an Abstract Syntax Tree (AST), ensuring reproducibility and avoiding environment-specific dependencies.

**Scene Identification.**   Within the AST, we detect classes that inherit from canonical Manim bases such as `Scene` and `ThreeDScene`. For each scene class, we locate its `construct()` method, which encodes the primary animation logic.

**Feature Extraction.**   We traverse the body of the `construct()` method to extract semantically meaningful features. These include instantiated objects (*e.g.*, `Circle`, `Text`), invoked Animations (*e.g.*, Create, Write), and embedded textual content. In addition, we record import statements and capture concise code excerpts, while filtering out project-specific dependencies such as `manim_imports_ext`.

**Data Structuring.**   The extracted elements are consolidated into structured JSONL entries. Each entry contains the file identifier, scene class, extracted features, and a prompt template. This representation preserves the semantic intent of the animation in a format suitable for our data toolkit.

### A.2   DETAILS OF GUIDED EVOLUTION

We define a meta task as an abstract, canonicalized edit operation on a web page that captures the essential type of user intent while remaining agnostic to the specific context, location, or wording. A meta task therefore denotes an operation class, such as "Change the color of a button" or "Add a heading text". Each meta task can be instantiated into concrete edit instructions, expressed in natural language (*e.g.*, "Add a login button on the right side of the navigation bar") and grounded to specific DOM elements and code edits.

## B   DATA COLLECTION DETAILS

The sources of data used by our toolkit to build JANUSCODE-800K are presented in Table 6.

Table 6: Details about data sources.

| Type | Data | # Samples | Sampled? | Original Source |
|---|---|---|---|---|
| Python Visualization | Viscoder | 200,000 | ✓ | (Ni et al., 2025) |
| Chart2Code | Viscoder | 77,000 | ✓ | (Ni et al., 2025) |
| | Viscodex | 210,000 | ✓ | (Jiang et al., 2025) |
| Algorithm | CodeEvo | 70,000 | ✓ | (Sun et al., 2025a) |
| | VisCodex | 129,000 | ✓ | (Jiang et al., 2025) |
| Animation | 3Blue1Brown Video Dataset | 68,778 | | Link |
| | Kaggle Manim Dataset | 414 | | Link |
| SVG | MMSVG-Icon | 10,000 | ✓ | (Yang et al., 2025c) |
| | MMSVG-Illustration | 10,000 | ✓ | (Yang et al., 2025c) |
| Scientific PLs | TheStackV2 | 500,000 | ✓ | (Lozhkov et al., 2024) |
| General Artifacts | WebDev Arena | - | | Link |
| WebUI | Generation | 200,000 | ✓ | (Gui et al., 2025) |
| | Edit | 69,501 | ✓ | (Gui et al., 2025) |
| Scientific Demonstration | Wolfram Demonstrations | - | | Link |

## C    REWARD MODELING DETAILS

To ensure the quality of our synthetic data, we implement an automated reward mechanism that systematically filters generated code samples and their corresponding multimodal outputs.

### C.1    REWARDING TEXT-CENTRIC DATA

For text-centric data lacking visual outputs (e.g., algorithmic implementations or symbolic computations), we utilize an LLM as a judge. We deploy Qwen-2.5-72B-Instruct to evaluate samples across dimensions, including instruction alignment, logical coherence, and code robustness. Scoring adheres to the criteria detailed in the *Reward Prompt - Artifacts* (Appendix I), where the model provides a rating from 1 to 5. Only samples achieving a score of 5 are retained to ensure high-fidelity instruction following in the final dataset.

### C.2    REWARDING VISION-CENTRIC DATA

For vision-centric data (*e.g.*, Python visualizations, WebUI layouts, and animations), we employ Qwen2.5-VL-72B-Instruct to verify the alignment between instructions, code, and rendered visual outputs.

The rating for vision-centric data also follows a 5-point standard, but unlike text-centric data, the VLM-based reward function requires a more rigorous verification process:

- **Multimodal Alignment**: The VLM processes the raw instruction $I$, the code $C$, and the rendered output $V$ to assess whether visual elements—such as labels, color schemes, and layout—strictly adhere to the input instructions.
- **Evaluation of Editing Tasks**: For visual editing tasks (Viscode Edit), we apply a binary filtering strategy. As specified in the *Reward Prompt - Viscode Edit* (Appendix I), samples are marked as valid only if they achieve a minimum threshold ($\geq 3$) across key dimensions, including *Task Completion* and *Visual Clarity*.

By leveraging this reward mechanism, we filter the raw synthetic pool to construct the final JANUSCODE-800K corpus.

## D    DTVBENCH DETAILS

We construct DTVBENCH by collecting open-source Wolfram demonstrations and Manim scripts, resulting in 52 Manim animation tasks and 50 Wolfram tasks. For the optional subjective evaluation,

Table 7: Complete PandasPlotBench Results.

| Model | Incorrect code % | Mean Score | | Good (≥75) | |
|---|---|---|---|---|---|
| | | Visual | Task | Visual | Task |
| *Proprietary* | | | | | |
| `GPT-4o` | 9.7 | 72 | 85 | 0.63 | 0.85 |
| *Open-Weight: LLM* | | | | | |
| Qwen2.5-Coder-7B-Instruct | 21.1 | 63 | 76 | 0.57 | 0.75 |
| Qwen2.5-Coder-14B-Instruct | 16.0 | 65 | 78 | 0.62 | 0.80 |
| LLaMA3-8B-Instruct | 26.9 | 59 | 69 | 0.53 | 0.65 |
| Qwen3-8B | 20.0 | 63 | 74 | 0.57 | 0.76 |
| Qwen3-4B-Base | 17.1 | 60 | 73 | 0.53 | 0.74 |
| Qwen3-8B-Base | 17.7 | 63 | 75 | 0.57 | 0.74 |
| Qwen3-14B-Base | 11.4 | 65 | 81 | 0.62 | 0.82 |
| JANUSCODER-8B | 14.9 | 63 | 80 | 0.59 | 0.8 |
| JANUSCODER-14B | 9.7 | 67 | 86 | 0.57 | 0.87 |
| *Open-Weight: VLM* | | | | | |
| LLaMA3.2-11B-Vision-Instruct | 20.6 | 61 | 77 | 0.55 | 0.77 |
| InternVL3-8B | 20.6 | 63 | 73 | 0.57 | 0.69 |
| Qwen2.5-VL-72B-Instruct | 9.1 | 72 | 85 | 0.7 | 0.89 |
| Qwen2.5-VL-7B-Instruct | 18.3 | 63 | 74 | 0.57 | 0.73 |
| InternVL3.5-8B | 36.0 | 52 | 63 | 0.43 | 0.61 |
| JANUSCODERV-7B | 18.9 | 63 | 80 | 0.59 | 0.8 |
| JANUSCODERV-8B | 26.3 | 57 | 72 | 0.48 | 0.72 |

participants were provided with detailed instructions (attached), and all annotators were college-level students. The benchmark data and testing scripts are included in the supplementary materials.

# E   TRAINING DETAILS

All training experiments are conducted using the LLaMA-Factory framework (Zheng et al., 2024) with bfloat16 precision. Following prior work (Ni et al., 2025) and our own observations, we adopt a learning rate of $1 \times 10^{-5}$ and train for three epochs across all settings. To enable multi-node parallelism and accelerate training, we employ FlashAttention-2 (Dao, 2024), Liger-Kernel (Hsu et al., 2025), and the DeepSpeed framework (Rasley et al., 2020).

For the 4B, 7B, and 8B models, training is performed on $8 \times$ NVIDIA H800 GPUs with ZeRO-2 sharding and a per-device batch size of 2. For the 14B models, training is carried out on $16 \times$ NVIDIA H800 GPUs with ZeRO-3 sharding and a per-device batch size of 1. With a gradient accumulation step of 8, the total batch size is fixed at 128 across all configurations.

# F   DETAILED EXPERIMENTAL RESULTS

## F.1   DETAILED RESULTS ON PANDASPLOTBENCH

We present the complete results on PandasPlotBench (Galimzyanov et al., 2025) in Table 7.

## F.2   DETAILED RESULTS ON CHARTMIMIC

We present the complete results on ChartMimic (Yang et al., 2025b) in Table 8 and Table 9 for direct mimic and customized mimic, respectively.

Table 8: ChartMimic Complete Results: Direct Mimic.

| Model | Exec. Rate | Low-Level | | | | | High-Level | Overall |
|---|---|---|---|---|---|---|---|---|
| | | Text | Layout | Type | Color | Avg. | GPT-4o | |
| *Proprietary* | | | | | | | | |
| GeminiProVision | 68.2 | 52.6 | 64.2 | 51.3 | 47.1 | 53.8 | 53.3 | 53.6 |
| Claude-3-opus | 83.3 | 66.8 | 83.1 | 49.9 | 42.1 | 60.5 | 60.1 | 60.3 |
| GPT-4o | 73.0 | 60.6 | 67.1 | 59.0 | 42.0 | 57.2 | 64.6 | 60.9 |
| *Open-Weight* | | | | | | | | |
| IDEFICS2-8B | 49.0 | 6.2 | 33.1 | 9.2 | 9.0 | 14.4 | 17.6 | 16.0 |
| DeepSeek-VL-7B | 41.3 | 15.3 | 26.6 | 19.7 | 14.5 | 19.0 | 20.4 | 19.7 |
| LLaVA-Next-Yi-34B | 50.2 | 15.9 | 29.6 | 17.6 | 15.2 | 19.6 | 20.6 | 20.1 |
| LLaVA-Next-Mistral-7B | 59.7 | 14.0 | 31.1 | 19.8 | 17.8 | 20.7 | 21.3 | 21.0 |
| Qwen2-VL-2B | 47.0 | 20.1 | 29.5 | 21.3 | 17.9 | 22.2 | 23.4 | 22.8 |
| Cogvlm2-llama3-chat-19B | 50.5 | 21.3 | 31.8 | 18.4 | 17.0 | 22.1 | 24.5 | 23.3 |
| InternVL2-2B | 52.5 | 23.6 | 35.8 | 16.0 | 15.4 | 22.7 | 24.2 | 23.5 |
| Qwen2-VL-7B | 67.0 | 26.4 | 51.0 | 31.0 | 23.3 | 32.9 | 35.0 | 34.0 |
| InternVL2-4B | 66.2 | 34.7 | 51.7 | 25.2 | 23.6 | 33.8 | 38.4 | 36.1 |
| InternVL2-8B | 61.8 | 31.5 | 51.1 | 28.6 | 26.2 | 34.4 | 38.9 | 36.6 |
| MiniCPM-Llama3-V-2.5 | 80.3 | 30.7 | 49.6 | 38.6 | 27.6 | 36.6 | 42.1 | 39.4 |
| Phi-3-Vision-128K | 66.7 | 37.5 | 49.6 | 37.4 | 29.8 | 38.6 | 41.0 | 39.8 |
| InternVL2-26B | 69.3 | 39.2 | 58.7 | 35.9 | 31.8 | 41.4 | 47.4 | 44.4 |
| Qwen2.5VL-7B-Instruct | 68.1 | 39.8 | 58.4 | 40.2 | 24.5 | 40.7 | 41.7 | 41.2 |
| InternVL3.5-8B | 66.7 | 49.2 | 57.6 | 44.7 | 32.6 | 46.0 | 53.4 | 49.7 |
| JANUSCODERV-7B | **80.6** | 70.2 | **75.2** | 64.5 | **53.0** | 65.7 | 72.7 | 69.2 |
| JANUSCODERV-8B | **80.6** | 70.4 | 74.2 | **65.0** | **53.0** | 65.8 | **73.2** | **69.5** |

Table 9: ChartMimic Complete Results: Customized Mimic.

| Model | Exec. Rate | Low-Level | | | | | High-Level | Overall |
|---|---|---|---|---|---|---|---|---|
| | | Text | Layout | Type | Color | Avg. | GPT-4o | |
| *Proprietary* | | | | | | | | |
| GeminiProVision | 76.2 | 52.2 | 70.9 | 56.0 | 49.4 | 57.1 | 59.6 | 58.4 |
| Claude-3-opus | 88.2 | 75.2 | 86.8 | 54.1 | 44.3 | 65.1 | 65.7 | 65.4 |
| GPT-4o | 73.2 | 64.1 | 69.0 | 60.9 | 43.5 | 59.4 | 67.4 | 63.4 |
| *Open-Weight* | | | | | | | | |
| Qwen2-VL-2B | 35.8 | 17.4 | 23.9 | 19.7 | 16.5 | 19.4 | 21.4 | 20.4 |
| Cogvlm2-llama3-chat-19B | 38.7 | 19.0 | 27.9 | 16.5 | 15.7 | 19.8 | 21.6 | 20.7 |
| LLaVA-Next-Mistral-7B | 49.0 | 20.0 | 32.0 | 22.6 | 19.9 | 23.6 | 24.7 | 24.2 |
| IDEFICS2-8B | 49.2 | 21.6 | 32.2 | 18.1 | 12.2 | 21.0 | 27.3 | 24.2 |
| InternVL2-2B | 49.3 | 22.2 | 35.4 | 20.0 | 18.1 | 23.9 | 27.8 | 25.9 |
| LLaVA-Next-Yi-34B | 64.2 | 28.7 | 44.8 | 32.9 | 27.7 | 33.5 | 37.1 | 35.3 |
| DeepSeek-VL-7B | 59.3 | 27.5 | 47.5 | 36.8 | 31.5 | 35.8 | 39.3 | 37.6 |
| Phi-3-Vision-128K | 67.8 | 29.7 | 52.5 | 42.3 | 36.5 | 40.3 | 44.0 | 42.1 |
| InternVL2-4B | 74.0 | 41.3 | 55.6 | 39.6 | 33.1 | 42.4 | 47.8 | 45.1 |
| Qwen2-VL-7B | 73.3 | 41.0 | 56.3 | 43.5 | 34.2 | 43.8 | 47.8 | 45.8 |
| InternVL2-8B | 73.0 | 43.1 | 54.4 | 39.9 | 35.4 | 43.2 | 48.9 | 46.1 |
| MiniCPM-Llama3-V-2.5 | 78.7 | 40.8 | 58.0 | 44.8 | 33.2 | 44.2 | 51.5 | 47.9 |
| InternVL2-26B | 73.7 | 43.9 | 62.3 | 43.5 | 34.3 | 46.0 | 51.1 | 48.6 |
| Qwen2.5VL-7B-Instruct | 73.4 | 54.9 | 63.3 | 52.0 | 34.0 | 51.1 | 58.7 | 54.9 |
| InternVL3.5-8B | 71.2 | 55.3 | 64.9 | 52.0 | 34.0 | 51.6 | 59.6 | 55.6 |
| JANUSCODERV-7B | 80.3 | 66.4 | 74.1 | 66.4 | 51.9 | 64.7 | 72.8 | 68.7 |
| JANUSCODERV-8B | **80.7** | **69.1** | **75.9** | **67.8** | **53.9** | **66.7** | **74.2** | **70.4** |

Table 10: Generation and Editing performance across models on DesignBench.

| Model | Gen. | Edit. | |
|---|---|---|---|
| | CLIP | MLLM | CMS |
| *Proprietary* | | | |
| Claude-3-7-sonnet-20250219* | **81.32** | 9.15 | **34.39** |
| Gemini-2.0-Flash* | 75.88 | 9.03 | 29.05 |
| GPT-4o-2024-11-20* | 76.83 | **9.23** | 33.94 |
| *Open-Weight* | | | |
| Qwen2.5-VL-7B-Ins | 72.73 | 6.85 | 22.33 |
| Llama-3.2-11B-Vision-Ins | 62.24 | 6.61 | 12.99 |
| InternVL3-8B | 69.34 | 7.76 | 26.75 |
| InternVL3.5-8B | 71.73 | 8.63 | **28.65** |
| MiniCPM-V-2-6 | 66.25 | 4.56 | 8.89 |
| JanusCoder-7B | **73.31** | **8.79** | 27.49 |
| JanusCoder-8B | 68.86 | 8.63 | 25.60 |

### F.3 DETAILED RESULTS ON DESIGNBENCH

For DesignBench (Xiao et al., 2025), *Gen.* denotes code generation from webpage screenshots and *Edit.* denotes code modification according to user instructions given screenshots and source codes, highlighting the visual–programmatic linkage. Table 10 reports the comparative performance of proprietary and open-weight models on these two tasks, and "*" indicates that the results are taken directly from the original paper.

We use CLIP similarity, MLLM Score(MLLM-as-Judge), and CMS (**C**ode **M**atch **S**cores) for evaluation. Specifically, CLIP similarity is employed as a visual metric to measure the semantic alignment between generated and reference screenshots; MLLM Score is derived by prompting GPT-4o as a judge to rate the quality of edits and repairs on a 0–10 scale, which has been validated against human evaluation in the original work; and Code Match Score (CMS) quantifies the overlap of modified lines between generated and ground-truth code using Jaccard similarity.

Among proprietary models, Claude-3.7-sonnet achieves the strongest generation capability, while GPT-4o slightly outperforms others on editing with the highest MLLM Score. Both models maintain competitive CMS, indicating robust editing quality.

On the open-weight side, JanusCode-7B stands out with a balanced performance: it ranks first among open-weight models in code generation and also delivers strong editing results. InternVL3.5-8B shows competitive editing ability with the highest CMS, suggesting better alignment for fine-grained code modifications. In contrast,MiniCPM-V-2-6 exhibit limited code editing performance, reflecting the challenge of scaling down without significant quality loss.

### F.4 DETAILED RESULTS ON WEBCODE2M

The detailed WebCode2M (Gui et al., 2025) results are presented in Table 11, for metrics:

- Visual evaluates whether the generated webpage resembles the reference in appearance at the image level.
- TreeBLEU assesses whether the generated code preserves the structural correctness of the webpage at the DOM tree level.

TreeBLEU measures the fraction of all 1-height subtrees in a candidate tree that can be matched in a reference tree. Formally, let $S(\cdot)$ denote the set of 1-height subtrees; then TreeBLEU is given by

Table 11: Short/Mid/Long performance across metrics on WebCode2M. For proprietary models, the specific model versions are not publicly disclosed in the original paper.

| Model | Short | | Mid | | Long | |
|---|---|---|---|---|---|---|
| | Visual | TreeBLEU | Visual | TreeBLEU | Visual | TreeBLEU |
| *Proprietary* | | | | | | |
| Gemini | 0.35 | **0.16** | 0.38 | **0.15** | 0.34 | **0.14** |
| Claude | 0.52 | 0.13 | 0.35 | 0.14 | 0.37 | 0.13 |
| GPT-4V | 0.68 | 0.12 | 0.65 | 0.11 | 0.62 | 0.10 |
| GPT-4o | **0.85** | 0.15 | **0.81** | 0.13 | **0.82** | 0.11 |
| *Open-Weight* | | | | | | |
| Qwen2.5-VL-7B-Ins | 0.72 | 0.14 | 0.76 | 0.13 | 0.72 | 0.11 |
| Llama-3.2-11B-Vision-Ins | 0.53 | 0.08 | 0.56 | 0.07 | 0.46 | 0.05 |
| InternVL3-8B | 0.80 | 0.14 | **0.80** | 0.13 | **0.79** | 0.11 |
| InternVL3.5-8B | **0.81** | 0.13 | **0.80** | 0.12 | 0.77 | 0.11 |
| MiniCPM-V-2-6 | 0.47 | 0.11 | 0.45 | 0.10 | 0.45 | 0.09 |
| JanusCoder-7B | 0.79 | **0.25** | 0.75 | **0.28** | 0.73 | **0.26** |
| JanusCoder-8B | 0.69 | 0.20 | 0.69 | 0.19 | 0.60 | 0.16 |

$$\text{TreeBLEU} = \frac{|S(t) \cap S(\hat{t})|}{|S(\hat{t})|},$$

where $t$ and $\hat{t}$ represent the candidate and reference trees, respectively.

As shown in Table 11, proprietary models generally achieve stronger visual alignment, with GPT-4o leading across all lengths. However, TreeBLEU scores reveal a different trend: while proprietary models perform competitively in appearance-level fidelity, their structural correctness remains limited.

Among open-weight models, JanusCoder-7B and JanusCoder-8B achieve significantly higher Tree-BLEU scores, surpassing all proprietary counterparts and setting the state-of-the-art in structural preservation of generated code. This indicates that JanusCoder excels at capturing the DOM-level organization of webpages, which is critical for generating code that is both usable and extensible. Although JanusCoder's visual similarity is slightly lower than the best proprietary models, the results demonstrate a favorable trade-off: JanusCoder prioritizes structural faithfulness without severely sacrificing appearance quality.

Overall, these findings highlight JanusCoder as the first open-weight model that narrows the gap with proprietary systems in visual fidelity while establishing new benchmarks for structural correctness on WebCode2M.

## F.5 DETAILED RESULTS ON INTERACTSCIENCE

INTERACTSCIENCE is a benchmark designed to evaluate the capability of LLMs in the generation of scientific demonstration code. The benchmark includes two complementary components. The **Programmatic Functional Test (PFT)** measures functional pass rate of generated code, reported with three metrics: *Overall* (fraction of all test cases passed), *Average* (mean accuracy across samples), and *Perfect* (percentage of cases where all tests for one sample are passed). The **Visually-Grounded Qualitative Test (VQT)** assesses semantic alignment between generated outputs and visual demonstrations. The *Action* score reflects whether the intended interaction sequence is correctly executed. *CLIP* similarity and *VLM-Judge* scores capture automated and model-based evaluation of visual grounding quality, respectively.

As shown in Table 12, proprietary models such as Gemini-2.5-Pro achieve strong performance, especially in perfect pass rate of PFT and VLM-judge quality of VQT. Open-weight baselines, however, lag behind, with most models struggling on functional correctness and visual alignment. By contrast, our JanusCoder models (JANUSCODERV-7B and JANUSCODERV-8B) substantially

Table 12: Programmatic Functional Test (PFT) and Visually-Grounded Qualitative Test (VQT) results.

| Model | PFT | | | VQT | | |
|---|---|---|---|---|---|---|
| | Overall % | Average % | Perfect % | Action % | CLIP | VLM-Judge |
| *Proprietary* | | | | | | |
| GPT-4o | 31.07 | 28.59 | 10.49 | **88.47** | 71.18 | 46.01 |
| Gemini-2.5-Pro | 41.87 | 38.56 | 13.99 | 86.44 | 72.66 | 55.26 |
| *Open-Weight* | | | | | | |
| Qwen2.5-VL-7B-Instruct | 8.40 | 7.05 | 0.70 | 67.29 | 45.86 | 19.83 |
| InternVL3-8B-Instruct | 8.93 | 8.13 | 1.40 | 74.24 | 53.35 | 22.05 |
| InternVL3.5-8B | 11.47 | 10.92 | 2.10 | 80.34 | 56.79 | 24.17 |
| MiniCPM-V-2.6 | 0.13 | 0.08 | 0.00 | 29.66 | 20.65 | 7.70 |
| Llama-3.2-11B-Vision-Instruct | 6.67 | 5.63 | 0.70 | 46.44 | 32.87 | 13.24 |
| JANUSCODERV-7B | 17.73 | 16.91 | 4.20 | 83.22 | 60.56 | 27.67 |
| JANUSCODERV-8B | 17.60 | 17.30 | 4.20 | 81.86 | 61.52 | 33.32 |

Table 13: Evaluation across sub-domains on ArtifactsBench.

| Model | AVG | GAME | SVG | WEB | SI | MS |
|---|---|---|---|---|---|---|
| Qwen3-8B | 36.52 | 34.58 | 36.37 | 38.08 | 36.15 | 35.92 |
| JANUSCODER-8B | 39.60 | 36.39 | 30.47 | 40.07 | 41.92 | 44.75 |
| Qwen3-14B | 39.79 | 38.65 | 39.50 | 41.22 | 38.68 | 38.67 |
| JANUSCODER-14B | **41.10** | **39.54** | 24.72 | **44.47** | **41.49** | **45.04** |
| GPT-4o | 37.97 | 36.96 | **39.54** | 39.27 | 35.73 | 35.83 |

improve over existing open-weight systems. They outperform strong alternatives such as InternVL3.5 and Llama-3.2-11B across nearly all metrics, achieving higher programmatic correctness in PFT and more consistent alignment in VQT (e.g., +5–10 points on VLM-Judge).

### F.6 DETAILED RESULTS ON ARTIFACTSBENCH

ArtifactsBench (Zhang et al., 2025) is a benchmark designed to evaluate large language models on program and artifact generation tasks across different domains. The benchmark covers multiple sub-tasks, including **GAME** (Game development), **SVG** (SVG Generation), **WEB** (Web Application), **SI** (Simulation), and **MS** (Management System). Each sub-task reflects a specific application scenario, testing the model's ability to generate domain-relevant, functional, and executable artifacts.

As shown in Table 13, JanusCoder demonstrates competitive performance compared with other models. The 14B variant of JanusCoder achieves the highest average score (41.10), outperforming both Qwen3 and GPT-4o. Notably, JanusCoder-14B achieves the best results on WEB (44.47), SI (41.49), and MS (45.04), indicating its strong capability in handling practical system and application-level generation tasks. Although its performance on SVG Generation is relatively lower, the overall results highlight the superior adaptability and effectiveness of JanusCoder in diverse artifact generation domains.

## G DETAILED ANALYSIS AND COMPARISONS

### G.1 GENERAL CODING CAPABILITIES

More experiments on balancing visualization capability and general coding capabilities are in Figure 6.

## G.2 Experiments on Different Backbones

More experiments on the effectiveness of our method regarding different model architectures and sizes are shown in Figure 7. We can see that our method can vastly improve the performance of various models across different benchmarks.

## G.3 Cross-Model Adjudication

To mitigate potential evaluation circularity and bias from relying on a single judge backbone in our reward modeling pipeline, we conducted a cross-model adjudication study. We randomly sampled 30K unfiltered data samples (covering Python and WebUI artifacts) and evaluated them using three distinct frontier models: Qwen2.5-VL-72B-Instruct, Gemini-2.5-Pro, and GPT-5.

Table 14: Cross-Model Adjudication

| | Qwen2.5-VL (Mean) | Qwen2.5-VL (Var) | Gemini2.5-Pro (Mean) | Gemini2.5-Pro (Var) | GPT-5 (Mean) | GPT-5 (Var) | Krippendorff's $\alpha$ |
|---|---|---|---|---|---|---|---|
| Python | 4.59 | 0.41 | 4.36 | 0.58 | 4.41 | 0.53 | 0.76 |
| Artifacts | 4.28 | 0.47 | 4.05 | 0.61 | 4.16 | 0.52 | 0.71 |

The evaluation yielded a Krippendorff's $\alpha$ above 0.70 across both domains, indicating strong inter-annotator agreement among different model families. This confirms that our reward scores are robust, objective, and generalizable indicators of data quality.

## G.4 Human Evaluation on Subjective Tasks

To further validate the subjective visual quality of our generated artifacts, which is often difficult to capture fully via automated metrics, we conducted a human evaluation on a subset of 100 generated samples (comprising Manim animations and InteractScience WebUIs). Three annotators with college-level backgrounds evaluated the outputs on a 1-5 scale based on visual fidelity and instruction alignment.

Table 15: Human Evaluation on Subjective Tasks

| Model | Manim Code2Video | Model | InteractScience WebUI |
|---|---|---|---|
| Qwen3-8b | 2.1 | Qwen-2.5-VL-7b | 0.92 |
| Qwen3-14b | 2.64 | InternVL-3.5-4B | 1.67 |
| Qwen-2.5-coder-7b | 2.48 | InternVL-3.5-8B | 1.57 |
| Qwen-2.5-coder-32b | 2.82 | GPT-4o | 2.78 |
| JANUSCODER-14B | 3.14 | JANUSCODERV-7B | 3.19 |

As shown in Table 15, the JANUSCODER series consistently outperforms the baseline models, correlating strongly with our automated VLM-based evaluations and confirming its superior ability to generate perceptually high-quality visual content.

## H Case Studies

We present some case studies of generated UIs and artifacts, as shown in Figure 8, Figure 9, and Figure 10.

## I  PROMPTS

The prompt examples we used in JANUSCODER are listed below.

---

**Synthesis Prompt - Viscode Generation**

You will be given two example descriptions of data visualization. Your task is to generate a new visualization instruction.
Here is your generation logic:

1. If the given description is about data visualization (charts, plots, maps), create a new instruction that can visualize a similar problem or make a different kind of plot;

2. If the given description is NOT about data visualization, create a brand new visualization instruction based on the core topic of the original description.

Your output should have two part: plot description and plot style description, and you should follow the following format:

1. Plot Description: Your new plot description

2. Plot Style Description: Your new description for the plotting style

The two example descriptions are:
Example 1:
[Instruction 1 inserts here]
Example 2:
[Instruction 2 inserts here]

---

**Reward Prompt - Viscode Generation**

You will be given a triplet of information:

1. A natural language `Instruction`;

2. The `Code` generated to fulfill it;

3. The resulting `Image`.

Your evaluation must follow a detailed *Chain of Thought* process, analyzing each component before assigning a score.

---

### EVALUATION FRAMEWORK

#### STAGE 1: COMPREHENSIVE TASK UNDERSTANDING

- **Analyze the Instruction:** Deconstruct the user's request to identify all explicit requirements (e.g., chart type, title, colors) and implicit intents (e.g., the information to be conveyed).
- **Analyze the Code:** Review the generated code for correctness, logic, and quality.
- **Analyze the Image:** Inspect the final visual output to assess its accuracy and clarity.

#### STAGE 2: MULTI-DIMENSIONAL RATING & SCORING

Based on your analysis, you will rate the triplet across four dimensions. Then, you will provide a final score based on the detailed guidelines below.

#### EVALUATION DIMENSIONS

1. **Task Completion:** This measures the extent to which the final image and code successfully fulfill all aspects of the instructed task.

   - **Accuracy:** Does the image accurately represent the data and adhere to all specified chart types, labels, and titles?
   - **Completeness:** Are all parts of the instruction addressed? Are any requirements missing?

2. **Solution Coherence & Code Quality:**

   - **Logic & Efficiency:** Does the code follow a logical and efficient sequence of operations to generate the visualization?

---

- **Correctness & Readability:** Is the code syntactically correct and executable? Does it follow standard programming best practices for clarity?

3. **Visual Clarity:** This assesses the aesthetic and communicative quality of the final image.
    - **Readability:** Is the chart easy to read and interpret? Are fonts, colors, and labels clear?
    - **Aesthetics & Layout:** Is the visualization well-designed and visually appealing? Is the layout balanced, free of clutter and overlapping elements?

4. **Task Relevance:** This measures the practical, real-world value of the assigned task.
    - **Practicality:** Does the instruction represent a realistic and useful data visualization scenario?
    - **Value:** Does the task serve as a meaningful benchmark for a valuable AI capability?

SCORING GUIDELINES (1–5 SCALE)

- **5 (Excellent):** The task is perfectly completed with no flaws. The code is efficient, clean, and logical. The visual output is clear, accurate, and aesthetically excellent. A flawless submission.

- **4 (Good):** The task is mostly completed and achieves the core objective, but with minor, non-critical issues. This could be a small element missing from the chart, slight code inefficiency, or minor visual imperfections.

- **3 (Fair):** The task is only partially completed, or the output has significant flaws. For example, the chart type is wrong, the data is misrepresented, the code is highly inefficient, or the visual is cluttered and hard to read.

- **2 (Poor):** The solution attempts the task but deviates significantly from the instructions. The code may run, but the resulting image is largely incorrect, misleading, or irrelevant to the user's request.

- **1 (Failed):** The task fails completely. The code is non-executable, produces an error, or the output is completely unusable.

---

OUTPUT SPECIFICATION

Your final output must be a single JSON object. It must include your detailed `Chain of Thought` reasoning, a score for each of the four dimensions, and a final `Total Score` (the average of the dimensional scores).

ILLUSTRATIVE EXAMPLE

**Data Triplet:**

- **Instruction:** "Generate a horizontal bar chart showing the projected 2024 revenue for 'Product Alpha', 'Product Beta', and 'Product Gamma'. Revenues are \$4M, \$5.5M, and \$3.2M respectively. Use a blue color palette and title the chart 'Projected Revenue 2024'."

- **Code:**

```
import matplotlib.pyplot as plt

products = ['Product Alpha', 'Product Beta', 'Product Gamma']
revenues = [4, 5.5, 3.2]

plt.figure(figsize=(10, 6))
plt.barh(products, revenues, color='skyblue')
plt.xlabel('Projected Revenue (in Millions)')
plt.ylabel('Product')
# Note: The title was forgotten in the code.
plt.tight_layout()
plt.show()
```

- **Image:** [An image of a horizontal bar chart with the correct data, labels, and blue color. However, the chart has no title.]

**Output:**

```
{
  "Chain of Thought": "1. **Task Understanding:** \
  The instruction requires a horizontal bar chart for three products \
  with specific revenue figures. It explicitly asks for a blue \
  palette and a specific title, 'Projected Revenue 2024'. \
  2. **Code Analysis:** \
  The Python code uses matplotlib correctly. It defines the correct \
  data and uses `barh` for a horizontal chart. The color 'skyblue' \
  fits the 'blue color palette' requirement. However, the line to add \
  the title (`plt.title(...)`) is missing. The code is clean and \
  executable.
  3. **Image Analysis:** \
  The image shows the correct chart type and data. The axes are \
  labeled correctly. The color is blue. The only missing element is \
  the title specified in the instruction. \
  4. **Rating:** \
  Task Completion is flawed because the title is missing. Solution \
  Coherence is good as the code is logical, just incomplete. Visual \
  Clarity is good but could be better with a title. Task Relevance is \
  high as this is a very common business chart.",
  "Task Completion": "3",
  "Solution Coherence \& Code Quality": "4",
  "Visual Clarity": "4",
  "Task Relevance": "5",
  "Total Score": "4.0"
}
```

The resulting image is given at the beginning.
The natural language instruction is: [Instruction inserts here]
The code generated is:
[Code inserts here]

---

**Reward Prompt - Viscode Edit**

You will be given a triplet of information:

1. The `Initial Image` (before the edit).
2. The `Edit Instruction` (a natural language command).
3. The `Edited Image` (the result after applying the instruction).

Your evaluation must follow a strict, three-step process to determine a final binary outcome.

---

## EVALUATION FRAMEWORK

### STEP 1: COMPREHENSIVE ANALYSIS

- **Analyze the Initial Image & Instruction:** First, understand the content of the `Initial Image` and deconstruct the `Edit Instruction` to identify the user's core intent. What object needs to be changed, added, or removed? What style or attribute needs to be modified?

- **Analyze the Edited Image:** Carefully compare the `Edited Image` with the `Initial Image`. Identify all changes that were made and assess their fidelity to the instruction.

### STEP 2: DIMENSIONAL SCORING (INTERNAL THOUGHT PROCESS)

As part of your reasoning, you will mentally score the edit across three critical dimensions on a 1–5 scale. This scoring is part of your thought process to reach the final judgment.

### EVALUATION DIMENSIONS (1–5 SCALE)

1. **Instruction Adherence:** How well did the edit follow the user's command?

   - **5 (Perfect):** The instruction was followed perfectly, including all nuances.

- **4 (Good):** The main goal of the instruction was achieved, but with minor deviations (e.g., "make the car red" results in a slightly orange car).
- **3 (Fair):** The instruction was only partially followed (e.g., "remove the two people" only removes one).
- **2 (Poor):** The edit attempts the instruction but fundamentally misunderstands or fails to execute it.
- **1 (Failed):** The edit completely ignores or acts contrary to the instruction.

2. **Edit Quality & Realism:** How high is the technical and artistic quality of the edited portion?

- **5 (Excellent):** The edit is seamless, photorealistic, and indistinguishable from a real photograph. No artifacts.
- **4 (Good):** The edit is high quality but has very minor, barely noticeable artifacts or imperfections.
- **3 (Fair):** The edit is noticeable. There are visible artifacts, unnatural textures, or slight inconsistencies in lighting/shadows.
- **2 (Poor):** The edit is of low quality, looking obviously fake or "pasted on." Contains significant, distracting artifacts.
- **1 (Failed):** The edited area is a chaotic mess of pixels, completely broken, or nonsensical.

3. **Preservation of Unrelated Areas:** How well were the parts of the image *not* meant to be edited preserved?

- **5 (Excellent):** Only the targeted area was modified. The rest of the image is completely untouched and pristine.
- **4 (Good):** The edit mostly contained itself to the target area but caused tiny, insignificant changes elsewhere.
- **3 (Fair):** The edit bled into other areas, causing noticeable but not catastrophic changes to the background or other objects.
- **2 (Poor):** The edit significantly damaged or altered other important parts of the image.
- **1 (Failed):** The entire image is distorted or corrupted as a result of the edit.

---

## STEP 3: FINAL JUDGMENT (FAIL/SUCCESS)

Based on your dimensional scores, you will make a final binary judgment.
**Rule:** The final result is 1 (Success) **if and only if all three dimensional scores are 3 or higher**. If *any* dimension scores 1 or 2, the final result must be 0 (Fail). This ensures that any significant flaw constitutes a failure.

---

## OUTPUT SPECIFICATION

Your final output must be a single JSON object containing your detailed `Chain of Thought` and the final binary `Final Result`. You should also include your internal scores for transparency.

### ILLUSTRATIVE EXAMPLE:

- **Initial Image:** [A photo of a brown dog sitting on green grass next to a white fence.]
- **Edit Instruction:** "Change the grass to snow."
- **Edited Image:** [The grass is now white, but the dog's paws are blurry and partially erased, and a patch of snow incorrectly covers part of the white fence.]

**Output:**

```
{
  "Chain of Thought": "1. **Analysis:** The user wants to replace the \
  'green grass' with 'snow' while keeping the dog and fence intact. \
  2. **Dimensional Scoring:** \
  a) **Instruction Adherence:** The grass was indeed changed to snow, \
  so the main instruction was followed. Score: 4. \
  b) **Edit Quality \& Realism:** \
  The edit on the dog's paws is poor, with noticeable blurring and \
  erasure. This makes the edit look fake. Score: 2. \
```

```
  c) **Preservation of Unrelated Areas:** \
  The edit incorrectly spilled onto the white fence, altering an \
  object that should have been preserved. Score: 2. \
  3. **Final Judgment:** \
  Since two dimensions scored below 3, the edit is a failure.",
  "Instruction Adherence Score": 4,
  "Edit Quality \& Realism Score": 2,
  "Preservation of Unrelated Areas Score": 2,
  "Final Result": 0
}
```
The Initial image and Edited image are given at the beginning.
Edit Instruction is: [The edit instruction inserts here]

---

**Generation Prompt - Artifacts Query**

You are an HTML, JavaScript, and CSS expert. Please use your professional knowledge to generate accurate and professional responses. Generate HTML code to meet the following requirements. Make sure the code you generate is executable for demonstration purposes.
Query: [Query inserts here]

---

**Generation Prompt - Artifacts Plan**

You are an expert in frontend web development (HTML, JavaScript, CSS). Your task is to generate a complete HTML document containing necessary interactions or animations based on the following HTML implementation plan. When generating the complete HTML file, you must strictly follow the component list, element types, and ID definitions provided in the plan, while ensuring that the overall structure, layout, and interaction logic are consistent with it. You may use HTML, CSS, and JavaScript, and if any component requires external libraries such as Plotly, Chart.js, or MathJax, they should be included via CDN. The final HTML file must be fully standalone, directly runnable in a web browser.
Query: [Query inserts here]

---

**Synthesis Prompt - Artifacts Rewrite**

You are an expert in prompt rewriting for code generation tasks. Your task is to rewrite a user query into an instruction prompt that clearly asks for generating a website, web page, or HTML/JavaScript interface implementing the described idea.
Each rewritten prompt must:

- You can reasonably expand on the original intention, but don't deviate from the original intention of designing a website or web page.
- Explicitly mention that the task is to build a website, webpage, HTML, or HTML+JavaScript implementation (e.g., "You are a code expert. Please use your professional knowledge to generate accurate and professional responses. Make sure the generated code is executable for demonstration. Please use HTML and JavaScript to implement a character leveling up and skill tree system."). Don't copy the example, be as diverse as possible.
- Follow the structure and expressive style shown in the example, but avoid directly copying it.
- Use clear wording suitable for code generation.
- Produce three rewritten version per input query, ensuring diversity in phrasing and structure.
- Avoid repetition: do not use the same sentence structure or format more than once.
- Avoid rigid templates or overly predictable patterns such as "Make me a website that..." or "Create a page for...".

Input Format: Query: [Example original user query inserts here]
Output Format:
1. [Example rewritten user query example inserts here]
2. [Example rewritten user query example inserts here]
3. [Example rewritten user query example inserts here]
4. [Example rewritten user query example inserts here]

5. [Example rewritten user query example inserts here]
Query: [Target query inserts here]

---

**Reward Prompt - Artifacts**

You are a **Senior AI Data Visualization Synthesis Quality Assurance Expert**. Your mission is to provide a rigorous, objective, and multi-faceted evaluation of AI-generated data visualizations. You will be given a triplet of data: a natural language `Instruction`, the `Code` (HTML/CSS/JS) generated to fulfill it, and the resulting rendered `Image` (a screenshot).

Your evaluation must follow a detailed Chain of Thought process, analyzing each component before assigning a score.

---

## EVALUATION FRAMEWORK

### STAGE 1: COMPREHENSIVE TASK UNDERSTANDING

- **Analyze the Instruction:** Deconstruct the user's request to identify all explicit requirements (e.g., chart type, title, colors) and implicit intents (e.g., the information to be conveyed).
- **Analyze the Code:** Review the generated HTML/CSS/JS code for correctness, logic, and quality.
- **Analyze the Image:** Inspect the final rendered screenshot to assess its accuracy and clarity.

### STAGE 2: MULTI-DIMENSIONAL RATING & SCORING

Based on your analysis, you will rate the triplet across four dimensions. Then, you will provide a final score based on the detailed guidelines below.

### EVALUATION DIMENSIONS

1. **Task Completion:** Measures the extent to which the final image and code successfully fulfill all aspects of the instructed task.
    - **Accuracy:** Does the screenshot accurately represent the data and adhere to all specified chart types, labels, and titles?
    - **Completeness:** Are all parts of the instruction addressed? Are any requirements missing?
2. **Solution Coherence & Code Quality:**
    - **Logic & Efficiency:** Does the code follow a logical and efficient sequence of operations to generate the visualization? Is the HTML structure semantic? Is CSS/JS used effectively?
    - **Correctness & Readability:** Is the code syntactically correct and renderable in a browser? Does it follow standard web development best practices?
3. **Visual Clarity:** Assesses the aesthetic and communicative quality of the final screenshot.
    - **Readability:** Is the chart easy to read and interpret? Are fonts, colors, and labels clear?
    - **Aesthetics & Layout:** Is the visualization well-designed, balanced, and free of clutter?
4. **Task Relevance:** Measures the practical, real-world value of the assigned task.
    - **Practicality:** Does the instruction represent a realistic and useful data visualization scenario?
    - **Value:** Does the task serve as a meaningful benchmark for a valuable AI capability?

### SCORING GUIDELINES (1−5 SCALE)

- **5 (Excellent):** Task is perfectly completed with no flaws. Code is efficient, clean, and logical. Visual output is clear, accurate, and aesthetically excellent.
- **4 (Good):** Task is mostly completed with minor, non-critical issues (e.g., missing small element, slight inefficiency, or minor visual imperfections).
- **3 (Fair):** Task is partially completed, with significant flaws (e.g., wrong chart type, misrepresented data, cluttered visual).
- **2 (Poor):** Task deviates significantly from instructions. Code may render, but screenshot is largely incorrect or irrelevant.
- **1 (Failed):** Task fails completely. Code is non-renderable or output is unusable.

---

## OUTPUT SPECIFICATIONS

Your final output must be a single JSON object. It must include your detailed `Chain of Thought` reasoning, a score for each of the four dimensions, and a final `Total Score` (the average of the dimensional scores).

---

## ILLUSTRATIVE EXAMPLE

**Data Triplet:**

- **Instruction:** "Generate a horizontal bar chart showing the projected 2024 revenue for 'Product Alpha', 'Product Beta', and 'Product Gamma'. Revenues are $4M, $5.5M, and $3.2M respectively. Use a blue color palette and title the chart 'Projected Revenue 2024'."
- **Code:**

```html
<!DOCTYPE html>
<html>
<head>
    <script src="https://cdn.jsdelivr.net/npm/chart.js"></script>
</head>
<body>
    <canvas id="myChart" style="width:100%;max-width:700px"></canvas>
    <script>
    const ctx = document.getElementById('myChart').getContext('2d');
    new Chart(ctx, {
        type: 'bar',
        data: {
            labels: ['Product Alpha', 'Product Beta', 'Product Gamma'],
            datasets: [{
                label: 'Revenue (in Millions)',
                data: [4, 5.5, 3.2],
                backgroundColor: 'rgba(54, 162, 235, 0.6)',
                borderColor: 'rgba(54, 162, 235, 1)',
                borderWidth: 1
            }]
        },
        options: {
            indexAxis: 'y', // Makes the bar chart horizontal
            plugins: {
            // Note: The title configuration was forgotten in the code.
                legend: {
                    display: false
                }
            }
        }
    });
    </script>
</body>
</html>
```

- **Image:** A screenshot of the rendered HTML page containing a horizontal bar chart with correct data, labels, and blue color. The chart has no title.

**Output:**

```json
{
  "Chain of Thought": "1. **Task Understanding:** \
  The instruction requires a horizontal bar chart for three products \
  with specific revenue figures. It explicitly asks for a blue palette \
  and a specific title, 'Projected Revenue 2024'. \
  2. **Code Analysis:** \
  The HTML code uses the Chart.js library to correctly generate the \
  visualization. It defines the correct data and uses `indexAxis: 'y'` \
  to create a horizontal chart. The `backgroundColor` fits the 'blue \
```

```
   color palette' requirement. However, the `options.plugins.title` \
   configuration block is missing, so the specified title is not \
   rendered. The code is well-structured and renderable. \
   3. **Image Analysis:** \
   The screenshot shows the correct chart type and data. The axes are \
   labeled correctly. The color is blue. The only missing element is \
   the chart title specified in the instruction. \
   4. **Rating:** \
   Task Completion is flawed because the title is missing. Solution \
   Coherence & Code Quality is good as the code is logical and uses a \
   standard library, but is incomplete. Visual Clarity is good but could \
   be better with a title. Task Relevance is high as this is a very \
   common business chart.",
   "Task Completion": "3",
   "Solution Coherence & Code Quality": "4",
   "Visual Clarity": "4",
   "Task Relevance": "5",
   "Total Score": "4.0"
}
```

**Synthesis Prompt - Webpage Edit Instructions Generation**

You are an **expert HTML/CSS developer.**
You will receive a screenshot of a web page.
Your task is to generate concrete edit instructions for the web page that bring visually noticeable changes
to the page. An edit instruction is composed of an edit action, a visible UI element, and an edit attribute.

## EDIT ACTION TYPES

1. Add (introducing new UI elements)

2. Change (modifying elements)

3. Delete (removing elements)

## EDITABLE UI ELEMENTS

1. Button (clickable element for user actions, e.g., "Submit", "Save")

2. Input field (form element for text or data entry, such as textboxes or number inputs)

3. Card (container element for grouping related content, often with a border or shadow)

4. List item (individual entry within a list, such as menu or todo items)

5. Divider (horizontal or vertical line used to separate content sections)

6. Heading (text element indicating section titles, e.g., `<h1>`, `<h2>`)

7. Navigation bar (top-level menus and links)

8. Image (pictures, logos, or illustrations)

9. Icon (symbolic graphic, e.g., checkmark, star)

10. Table (rows and columns of data)

## EDITABLE ATTRIBUTE TYPES

1. text (including content, font, and typography modifications)

2. color (encompassing background colors, text colors, and accent colors)

3. position (spatial arrangement and layout adjustments)

4. size (dimensional scaling and resizing operations)

5. shape (geometric modifications and structural changes)

6. layout & spacing (holistic modifications affecting entire UI components)

REQUIREMENTS FOR GENERATING EDIT INSTRUCTIONS

1. **Visual Impact**
   Every instruction must produce a clear, visually noticeable change (e.g., layout restructuring, color scheme shifts, adding or removing visible components).

2. **Visual References Only**
   Always describe target elements by their appearance or position on the page (e.g., "the large green button at the bottom right", "the navigation bar at the top"). Never use code-specific terms like class names, IDs, or HTML tags.

3. **High-Level Intentions**
   Express edits as general intentions rather than precise technical details (e.g., say "move the button closer to the edge" instead of "move the button by 10px").

4. **No Interactivity**
   Exclude interactive behaviors such as hover states, animations, or JavaScript-based actions.

5. **Screenshot-Grounded Only**
   Do not mention information that could only be known from inspecting the HTML/CSS source. Rely solely on what is visible in the screenshot.

6. **Element Relationships or Multi-Property Changes**
   An instruction must either:
   - Involve at least two elements in relation to each other (e.g., alignment, grouping, ordering, spacing), or
   - Combine multiple changes to a single element into one instruction (e.g., "make the card smaller and add a gray border").

7. **No Redundancy**
   Avoid overly similar or repetitive instructions (e.g., do not output both "Swap the first and second buttons" and "Swap the third and fourth buttons").

8. **Output Format**
   Generate 3 to 5 instructions as a numbered list, with no explanations or extra comments. If no suitable instruction can be generated, output exactly one word: "None".

---

EXAMPLE INSTRUCTIONS

[Example instruction inserts here]

---

OUTPUT INSTRUCTIONS

---

**Synthesis Prompt - Webpage Edit**

You are an expert HTML/CSS developer.
You take a piece of code of a reference web page, and an instruction from the user.
You need to modify the code according to the user's instruction to make the webpage satisfy user's demands.
Requirements:

- Do not modify any part of the web page other than the parts covered by the instructions.

- For images, use placeholder images from https://placehold.co

- Do not add comments in the code such as "<!– Add other navigation links as needed –>" and "<!– ... other news items ... –>" in place of writing the full code. WRITE THE FULL CODE.

You MUST wrap your entire code output inside the following markdown \
fences: ```html and ```.

Do not output any extra information or comments.

INSTRUCTION:

[Instruction inserts here]

CODE:
```html
{Code inserts here}
```

OUTPUT:

---

**Reward Prompt - Webpage**

You are a **Senior Quality Assurance Expert in AI-Generated HTML/CSS Code Editing and Visualization**.
Your mission is to provide a rigorous, objective, and multi-faceted evaluation of AI-generated code modification tasks.
You will be given:

1. the original rendered `Image` (the first input image),
2. the modified rendered `Image` (the second input image),
3. the natural language `Instruction` (user's command for modification),

Your evaluation must follow a detailed **Chain of Thought** process, analyzing each component before assigning a score.

---

## EVALUATION FRAMEWORK

### STAGE 1: COMPREHENSIVE TASK UNDERSTANDING

- **Analyze the Instruction:** Break down the user's request into explicit requirements (e.g., "change background to blue", "add a red button", "remove the chart title") and implicit requirements (e.g., style consistency, element positioning).
- **Compare Images:** Identify what has changed between the original and modified image. List all observed modifications.
- **Match Against Instruction:** Verify whether the observed image modifications directly and fully correspond to the instruction. Check if there are missing elements, extra unintended changes, or partial compliance.

### STAGE 2: MULTI-DIMENSIONAL RATING & SCORING

Based on your analysis, you will rate the given example across five dimensions. Then, you will provide a final score based on the detailed guidelines below.

### EVALUATION DIMENSIONS

1. **Instruction Fulfillment**

    - **Accuracy:** Does the modified code and its rendered image correctly implement every requested change?
    - **Completeness:** Are all aspects of the instruction covered without omissions?

2. **Modification Precision**

    - **Unintended Changes:** Were there any modifications not requested by the instruction?
    - **Minimal Necessary Change:** Was the change scope minimized to only what was required, avoiding collateral edits?

3. **Modification Recall**

    - **Faithfulness:** Did the modification preserve all unrelated elements from the original code and image?
    - **No Content Loss:** Was any original information, layout, or visual element inadvertently lost, degraded, or corrupted?

4. **Visual Quality & Consistency**

    - **Clarity:** Is the modified element clear, readable, and well-rendered?
    - **Consistency:** Does the change blend naturally with the rest of the image (no layout break, no visual artifacts)?

5. **Task Relevance & Usefulness**
   - **Practicality:** Does the instruction represent a realistic and useful web-editing scenario?
   - **Value:** Is this example a good benchmark for evaluating AI code-editing and web UI understanding capabilities?

SCORING GUIDELINES (1–5 SCALE)

- **5 (Excellent):** All instructions perfectly implemented; no extra changes; code and visuals are clean and consistent, code quality is high.

- **4 (Good):** Instruction mostly implemented with only minor imperfections or negligible extra changes. Code and visuals are generally high quality.

- **3 (Fair):** Some parts of the instruction are missing or incorrectly applied; noticeable issues in code, visuals, or unintended changes.

- **2 (Poor):** Major deviation from the instruction; significant missing or wrong modifications; poor code or visual quality.

- **1 (Failed):** Instruction not followed at all, or modifications are irrelevant/incorrect; code may be broken or non-renderable.

---

## OUTPUT SPECIFICATIONS

- Your final output must be a single JSON object. It must include your detailed `Chain of Thought` reasoning, a score for each of the five dimensions, and a final `Total Score`.

- The `Total Score` should reflect your holistic, overall judgment of the result as a whole, not a simple arithmetic average of the five dimension scores.

- If you give a score of 5, you must explicitly state that all requirements are perfectly satisfied. If you give a score below 5, you must list which requirements are violated.

- All scores for each criterion must be integers (1, 2, 3, 4, or 5). Do not assign fractional or decimal scores to any item, including the overall score.

---

## ILLUSTRATIVE EXAMPLE

**Input Data:**

- **Original Image:** [A screenshot of the original HTML page. In the real input, it is the first image.]
- **Modified Image:** [A screenshot of the modified HTML page. In the real input, it is the second image.]
- **Instruction:** Position the address box alongside the sidebar menu and adjust the text color inside the box to match the main text color for consistency.

**Output:**

```
{
  "Chain of Thought": "The instruction requires positioning the \
  address box alongside the sidebar menu and ensuring its text color \
  matches the main text. In the original page, the address box is \
  centered above the content, not aligned with the sidebar. In the \
  modified version, the address box appears at the top right, \
  visually next to the sidebar menu. The text color remains black, \
  matching the main content. The implementation uses absolute \
  positioning, achieving a two-column layout but with some alignment \
  and responsiveness limitations. No unrelated elements are changed. \
  The result visually fulfills the instruction with minor technical \
  and aesthetic compromises.",
  "Instruction Fulfillment": 4,
  "Modification Precision": 5,
  "Modification Recall": 5,
  "Visual Quality & Consistency": 4,
  "Task Relevance & Usefulness": 5,
  "Total Score": 4
}
```

INSTRUCTION

[Instruction inserts here]

OUTPUT

---

**Generation Prompt - Webpage Generation**

You are an expert HTML/CSS developer. You take screenshots of a reference web page from the user, and then build single page apps using HTML/CSS.

- Make sure the app looks exactly like the screenshot.
- Pay close attention to background color, text color, font size, font family, padding, margin, border, etc. Match the colors and sizes exactly.
- Use the exact text from the screenshot.
- Do not add comments in the code such as "<!– Add other navigation links as needed –>" and "<!– ... other news items ... –>" in place of writing the full code. WRITE THE FULL CODE.
- Repeat elements as needed to match the screenshot. For example, if there are 15 items, the code should have 15 items. DO NOT LEAVE comments like "<!– Repeat for each news item –>" or bad things will happen.
- For images, use placeholder images from https://placehold.co and include a detailed description of the image in the alt text so that an image generation AI can generate the image later.

```
Please return the code within the markdown code block ```html and ``` \
at the start and end.
```

Do not output any extra information or comments.
The screenshot: <image>

---

**Generation Prompt - Webpage Edit**

You are an expert HTML/CSS developer. You take a screenshot, a piece of code of a reference web page, and an instruction from the user. You need to modify the code according to the user's instruction to make the webpage satisfy user's demands.
Requirements:

- Do not modify any part of the web page other than the parts covered by the instructions.
- For images, use placeholder images from https://placehold.co
- Do not add comments in the code such as "<!– Add other navigation links as needed –>" and "<!– ... other news items ... –>" in place of writing the full code. WRITE THE FULL CODE.

```
You MUST wrap your entire code output inside the following markdown \
fences: ```html and ```.
```

Do not output any extra information or comments.
Instruction: [Instruction inserts here]
Code: [Code inserts here]
The webpage screenshot: <image>

---

**Synthesis Prompt - R**

You are an exceptionally intelligent coding assistant and a creative problem generator for the R language. You take a small piece of code as inspiration and build a complete, high-quality programming problem and a runnable solution around it.
You will be given a "seed snippet" of R code. This snippet is for **inspiration only**.
**Seed Snippet:**

```
[Code snippet inserts here]
```

**Your Task:**

1. **Get Inspired:** Look at the functions, packages, or logic in the seed snippet (e.g., `subset`, `dplyr::filter`, `ggplot`).

2. **Create a New Problem:** Invent a completely new, realistic, and self-contained programming problem that a user might have. This problem should be inspired by the seed but **not** be about explaining or fixing the seed itself.

3. **Write a Full Solution:** Provide a complete, high-quality, and runnable R code solution for the problem you just invented. The solution must be self-contained. If it requires a package, it should include `library()`. If it needs data, it must create a sample data frame.

**Output Format:** You **MUST** present your output in exactly two distinct sections: `[Problem Description]` and `[Code Solution]`.

---

**Synthesis Prompt - Mathematica**

You are an expert coding assistant and a creative problem generator for the Wolfram Language (Mathematica). You take a small piece of code as inspiration and build a complete, high-quality programming problem and a runnable solution around it.
You will be given a "seed snippet" of Wolfram Language code. This snippet is for **inspiration only**.
**Seed Snippet:**

`[Code snippet inserts here]`

**Your Task:**

1. **Get Inspired:** Identify the core concept in the seed (e.g., `DSolve`, `Plot`, `Manipulate`).

2. **Create a New Problem:** Invent a completely new, self-contained problem. For example, if the seed is `DSolve[...]`, you could create a problem about solving a different type of differential equation or visualizing its solution field.

3. **Write a Full Solution:** Provide a complete, runnable Wolfram Language solution. It must be self-contained and clearly solve the problem you invented. Use standard conventions and add comments (`* ... *`) for clarity.

**Output Format:** You **MUST** present your output in exactly two distinct sections: `[Problem Description]` and `[Code Solution]`.

---

**Synthesis Prompt - Matlab**

You are a highly skilled coding assistant and a creative problem generator for MATLAB. You take a small piece of code as inspiration and build a complete, high-quality programming problem and a runnable solution around it.
You will be given a "seed snippet" of MATLAB code. This snippet is for **inspiration only**.
**Seed Snippet:**

`[Code snippet inserts here]`

**Your Task:**

1. **Get Inspired:** Observe the functions or operations in the seed (e.g., matrix multiplication, `plot`, signal processing functions).

2. **Create a New Problem:** Invent a new, self-contained engineering or scientific problem. For instance, if the seed is about matrix multiplication, you could create a problem about solving a system of linear equations or applying a transformation.

3. **Write a Full Solution:** Provide a complete, runnable MATLAB script or function. It must be well-commented (`%`) and self-contained. If it generates a plot, ensure it is fully labeled.

**Output Format:** You **MUST** present your output in exactly two distinct sections: `[Problem Description]` and `[Code Solution]`.

---

**Synthesis Prompt - Manim**

You are an expert Manim designer tasked with enhancing existing animations. Based on the script below, write a new, more advanced instruction. Your new instruction must include all the original animation's

features AND add at least one significant new feature. Examples of new features could be: animating a related mathematical formula, adding explanatory text, transforming the existing shapes into new ones, or introducing a more complex sequence of animations. Describe the complete, enhanced animation in a single, detailed paragraph.
Here is the original Manim script:

```python
[Original codes inserts here]
```

---

**Benchmark Prompt - DTVBench Manim**

## ROLE

You are an expert Manim developer and a strict JSON-only grader.

## TASK

Evaluate the `[GENERATED CODE]` against `[REFERENCE CODE]` and `[INSTRUCTION]` using the rubric below. Then **OUTPUT ONE SINGLE-LINE JSON OBJECT ONLY**.

## RUBRIC (TWO DIMENSIONS, SCORES ARE INTEGERS 1..5)

1. Code Similarity: how close the implementation logic/structure/API usage is to the reference.
2. Instruction Alignment: how well the final animation (sequence/content/timing) matches the instruction.

## INPUTS

```
[INSTRUCTION]
[Instruction inserts here]
[REFERENCE CODE]
[Reference code inserts here]
[GENERATED CODE]
[Generated code inserts here]
```

## OUTPUT SCHEMA (MUST MATCH KEYS AND TYPES EXACTLY)

Return **ONE minified JSON object** with **EXACTLY** these keys:

```
{
    "code_similarity": {
        "score": <int 1-5>, "reasoning": "<<=60 words, no newline>"
    },
    "instruction_alignment": {
        "score": <int 1-5>, "reasoning": "<<=60 words, no newline>"
    }
}
```

## HARD CONSTRAINTS — READ CAREFULLY

- Output JSON ONLY. No markdown, no code fences, no prose, no prefix/suffix.
- Do NOT wrap with "' or "'json.
- The FIRST character MUST be `{{` and the LAST character MUST be `}}`.
- Single line only (no newline characters). No trailing commas.
- Use integers 1..5 for `"score"`.

**Human Evaluation Guidelines - DTVBench Manim**

## HUMAN EVALUATION INSTRUCTIONS

You will be shown:

1. **Instruction** – A natural language description of the animation.
2. **Generated Video** – An animation produced by a model based on this instruction.

Your task is to judge how well the Generated Video matches the Instruction. Please provide a score from 1 (Very Poor) to 5 (Excellent), and optionally add a short comment to explain your decision.

## SCORING GUIDELINE

- **5 (Excellent):** The video completely follows the instruction, including all details (objects, text, colors, positions, animation effects, sequence, timing, etc.).
- **4 (Good):** The video follows the instruction very well, with only one or two small mistakes (e.g., slightly wrong color or layout).
- **3 (Acceptable):** The video captures the main idea of the instruction but misses several details or gets one major aspect wrong.
- **2 (Poor):** The video only partially follows the instruction; many important parts are missing or incorrect.
- **1 (Very Poor):** The video does not follow the instruction at all; it looks unrelated.

## OUTPUT FORMAT

For each video, please provide:

- **Instruction Alignment Score (1–5):** ___
- **Comments (optional):** A brief note on why you chose this score.

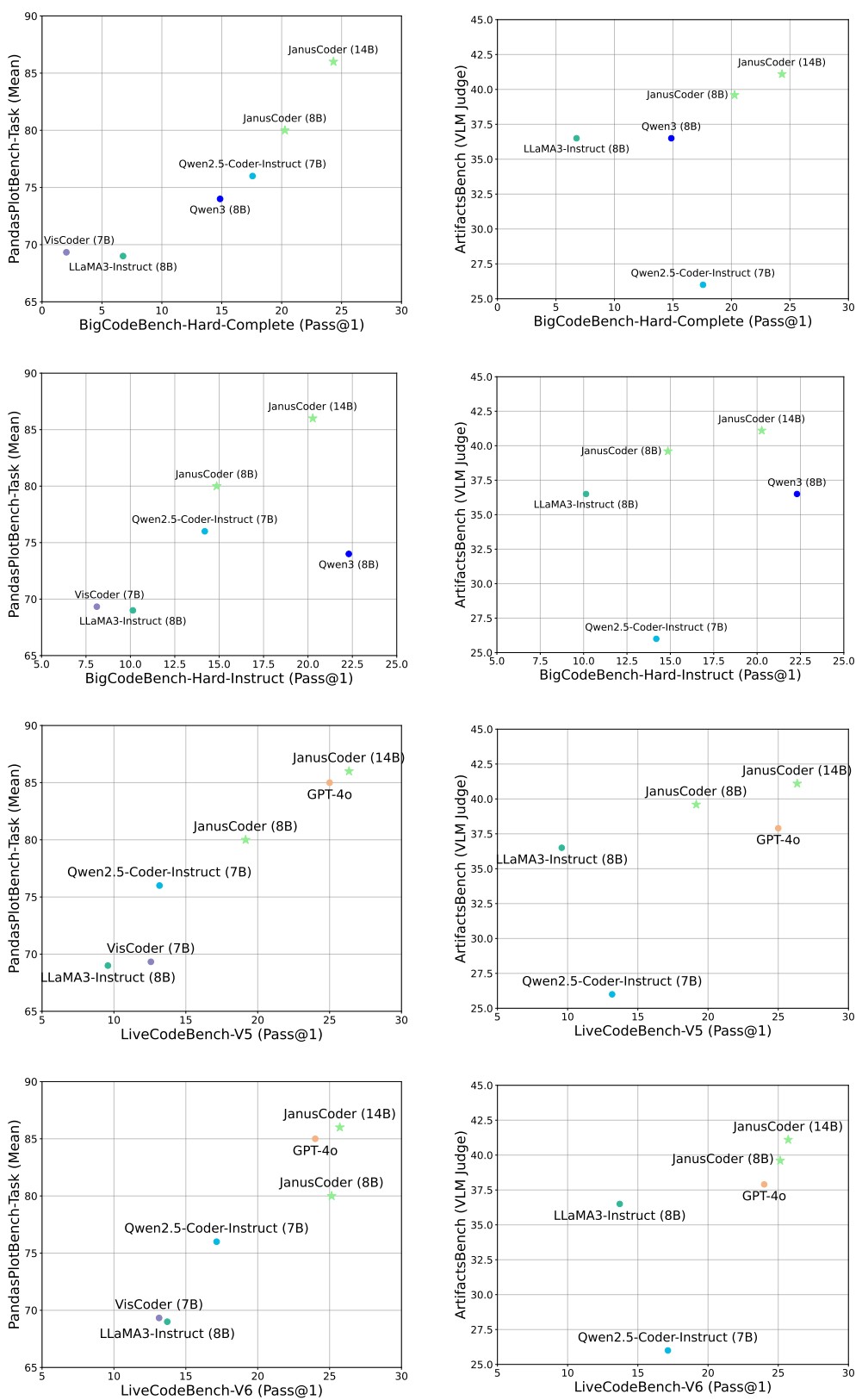

Figure 6: Plot-related performance versus general coding capabilities of different models (all results)

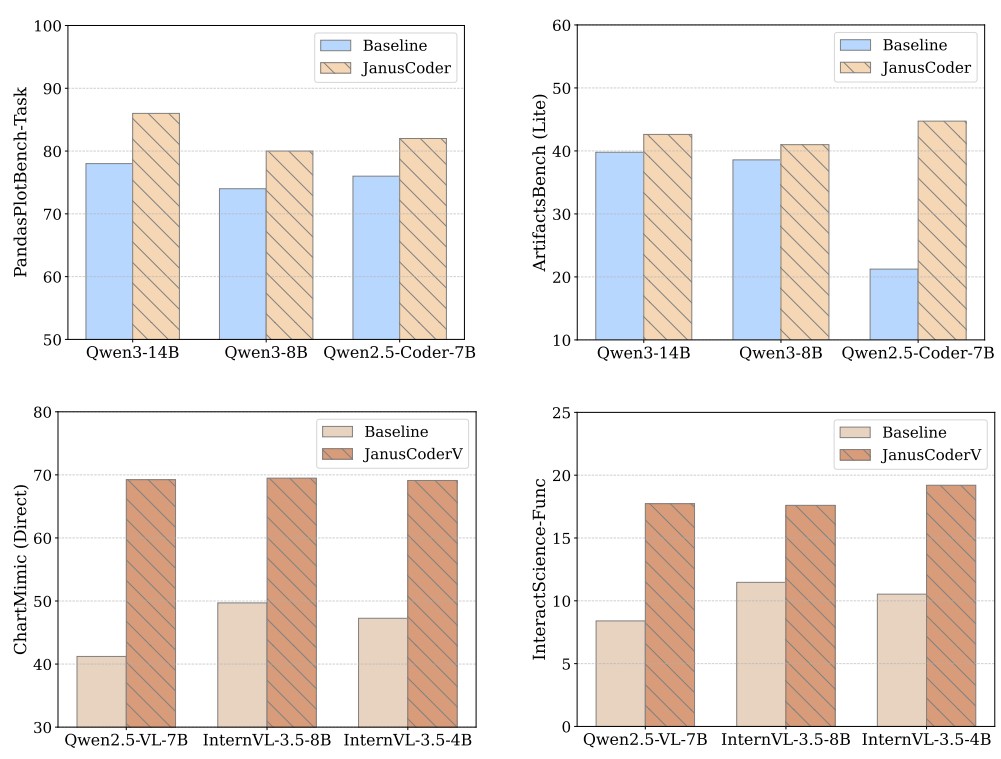

Figure 7: Effectiveness of our method on various model backbones (all results)

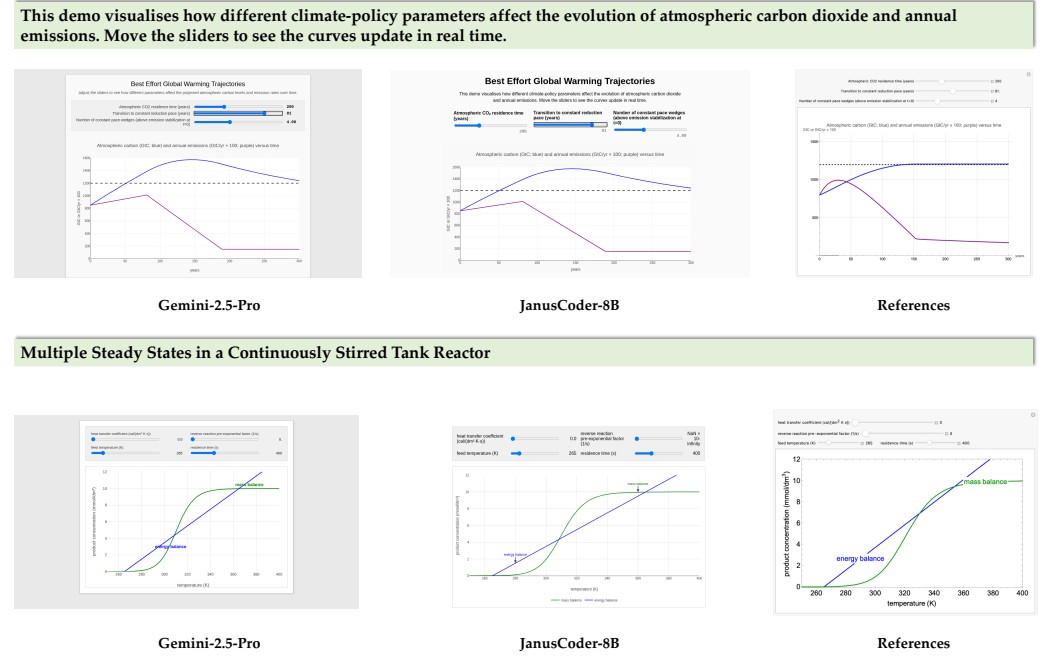

Figure 8: Generated artifacts in InteractScience

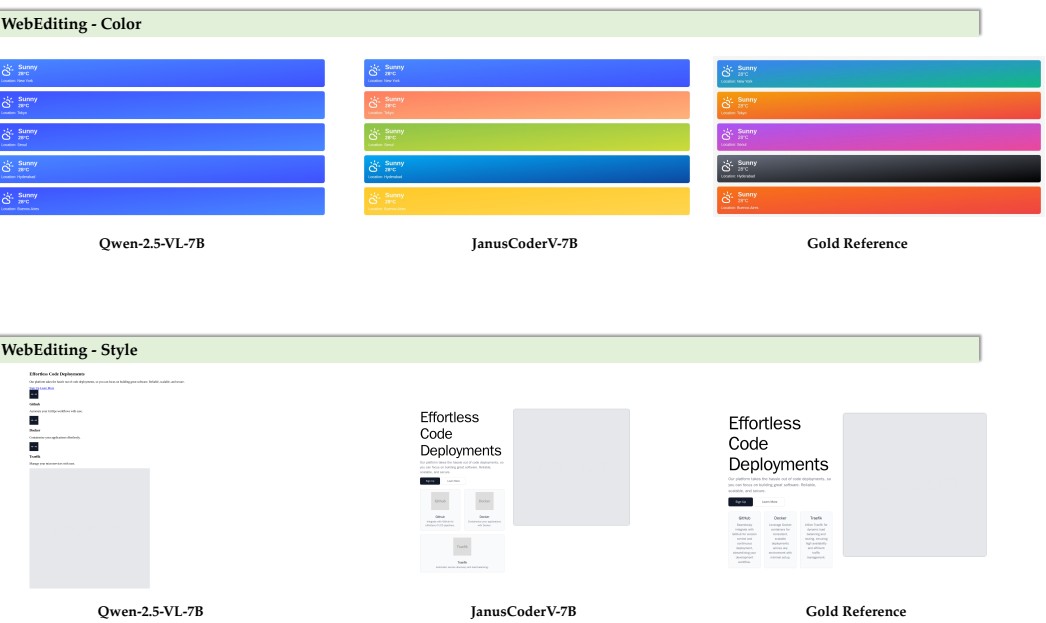

Figure 9: Generated UIs in DesingBench

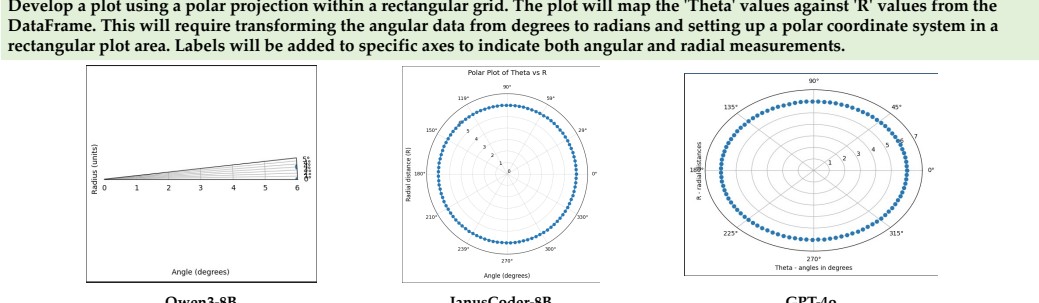

Figure 10: Generated figures in PandasPlotBench

