# OpenReview forum: "JanusCoder: Towards a Foundational Visual-Programmatic Interface for Code Intelligence"
_ICLR.cc/2026/Conference — ICLR 2026 Poster_

### Official Review · Reviewer_uHBB · 2025-10-31

**Soundness:** 4
**Presentation:** 4
**Contribution:** 4
**Rating:** 8
**Confidence:** 5

**Summary:**

This paper introduces JanusCoder, a family of multimodal code-generation models that integrate textual, visual, and programmatic information. Its key contributions include:
1.  JanusCode-800K Dataset: A new dataset covering both text-centric and vision-centric tasks;
2.  DTVBench Benchmark: A new benchmark for dynamic theorem visualization using Manim and Mathematica;
3.  Strong Performance: Comprehensive evaluations on eight benchmarks show JanusCoder performs on par with or surpasses GPT-4o and specialized open-source models.

**Strengths:**

1.  This paper introduces a comprehensive dataset (JanusCode-800K), which fills a clear gap by integrating visual, textual, and programmatic modalities at scale;
2. The paper is well-written, with clear motivation and structured sections to detail the data curation process;
3. JanusCoder obtains strong empirical results across >8 benchmarks;

**Weaknesses:**

1. DTVBench’s limited scale (~102 tasks) may constrain statistical reliability;
2. Why does JANUSCODERV-8B perform worse than InternVL3.5-8B on DesignBench and WebCode2M?
3. When considering visual information, besides benchmarks purely focused on visualization, there are also some algorithmic or reasoning-related benchmarks [1]. Has the paper evaluated or discussed model performance on such tasks?
4. A question mark appears at line 253.


[1] MMCode: Benchmarking Multimodal Large Language Models for Code Generation with Visually Rich Programming Problems

**Questions:**

See the Weakness part.

---

> ### Author Response · Authors · 2025-11-18
> **Response to Reviewer uHBB**
>
> Thanks for your review! We appreciate the positive support for our data / model contributions, the strong empirical results, and the paper's presentation. We summarize and address the questions raised as follows:
>
> > W1: The scale of DTVBench
>
> We position DTVBench as an auxiliary contribution, representing a pioneering effort to evaluate complex code generation (e.g., interactive Mathematica/Manim) where no established benchmarks existed when conducting this work
> The scale is limited by: (1) the scarcity of public complex theorem visualization data and (2) copyright restrictions.
>
> We therefore focused on quality and robustness over quantity. All tasks were manually curated and verified from professional sources (e.g., Wolfram demos, 3Blue1Brown public source codes) to ensure correctness. We expect future, larger benchmarks in this domain and will gladly evaluate them when available.
>
> > W2: JanusCoderV-8B's performance on DesignBench and WebCode2M
>
> Thank you for this detailed observation. We briefly discuss this phenomenon in Appendix E.4.
>
> First, while InternVL-3.5 has a higher visual similarity score, JanusCoderV achieves a significantly higher TreeBLEU score (code similarity). Given our data pipeline, our model prioritizes generating functional code, which is critical for usability. This focus on structural correctness (discussed in E.4) can sometimes come at the expense of pixel-level visual similarity.
>
> Further, JanusCoderV is a unified model trained on a highly diverse range of tasks. It is possible that the 8B model scale limits its capacity, leading to some forgetting of web-specific data from the post-training phase. Given our strong performance on all other benchmarks, we hypothesize this is a capacity limitation that could be resolved by scaling the backbone model.
>
> > W3: Has the paper evaluated or discussed model performance on algorithmic or reasoning-related benchmarks?
>
> Thank you for this question. Yes, we did evaluate our models on tasks that integrate visual coding with algorithms and reasoning.
> - For algorithms: We validated our model's core algorithmic strength on LiveCodeBench [1] and BigCodeBench [2] (Sec 6.3 and Appendix F.1 General Coding Capabilities). These results (presented in our visualizaed analysis) confirm its robust and balanced capabilities across both complex visualization and general-purpose algorithmic tasks. We will add a sentence to the main text to make this connection clearer.
> - For reasoning: Our theorem generation tasks inherently require this. For example, the InteractScience [3] (Sec 5.4) benchmark requires the model to first reason for a scientific theorem and then generate code for its interactive demonstration, explicitly fusing visual understanding with reasoning.
>
> We already cited the paper you suggested in our current version. We note that this paper does not yet include results for recent opensource / commercial models, and supplementing these baseline experiments would require a longer timeframe and resources, which we will consider later.
>
> Moreover, we thank you for pointing out (W4) the formatting issue (the question mark at line 253 was an indexing error). This will be corrected.
>
> Finally, we thank you again for your recognition and support of our work.
>
> ### Reference
>
> [1] LiveCodeBench: Holistic and Contamination Free Evaluation of Large Language Models for Code https://arxiv.org/abs/2403.07974
>
> [2] BigCodeBench: Benchmarking Code Generation with Diverse Function Calls and Complex Instructions https://openreview.net/forum?id=YrycTjllL0
>
> [3] InteractScience: Programmatic and Visually-Grounded Evaluation of Interactive Scientific Demonstration Code Generation https://arxiv.org/abs/2510.09724

---

### Official Review · Reviewer_NYgb · 2025-11-01

**Soundness:** 3
**Presentation:** 3
**Contribution:** 3
**Rating:** 6
**Confidence:** 5

**Summary:**

JanusCoder expands code intelligence from purely text-based inputs to jointly reasoning over code and visual outputs. It introduces a scalable multimodal code-data synthesis pipeline and builds JANUSCODE-800K, the largest multimodal code corpus to date. The authors develop unified models that handle text-centric and vision-centric coding tasks, achieving performance comparable to or surpassing commercial systems while offering insights into aligning program logic with visual expression.

**Strengths:**

1. The paper introduces a unified visual-programmatic interface and a complete multimodal code data synthesis toolkit, enabling code models to reason jointly over textual and visual programming tasks.

2. The work builds JANUSCODE-800K, the largest multimodal code corpus, and conducts extensive experiments across diverse benchmarks and modalities, demonstrating careful design and thorough empirical evaluation.

3. Results show strong performance, with 7B, 14B models approaching or surpassing commercial systems across text-centric and vision-centric code tasks.

**Weaknesses:**

1. Unclear data release plan: The data synthesis pipeline and JANUSCODE-800K corpus are central contributions, yet public availability is not guaranteed at submission time, raising concerns about reproducibility and community impact. The paper should explicitly clarify the dataset release schedule and scope.

2. Judge-based evaluation bias: The Stage-3 refinement heavily depends on LLM/VLM judges, which risks evaluation circularity and bias toward the same models used for filtering. More human evaluation or cross-model adjudication would strengthen reliability and reduce bias.

3. Limited methodological novelty: Despite strong engineering effort, the paper reads largely as a dataset and data-pipeline contribution. The model setup closely follows existing architectures and training paradigms, making the work feel more like a large-scale data release than a method advance.

**Questions:**

JANUSCODE-800K is central to the contribution. Will the full dataset be publicly released at camera-ready?

---

> ### Author Response · Authors · 2025-11-18
> **Response to Reviewer NYgb**
>
> Thank you for your review! We appreciate your positive feedback on the scope and performance of our model. We address your questions as follows:
>
> > W1 & Q1: Data release plan. Will the full dataset be publicly released at camera-ready?
>
> Yes. We have already publicly released a portion of the data (which could not be included in this submission as it exceeded OpenReview's supplementary file size limit). We are currently finalizing the process to comply with company data regulation policies and expect to release the full dataset before the camera-ready deadline.
>
> Furthermore, we release the code of our data synthesis toolkit. This allows other researchers to leverage our pipeline with more powerful future models to generate even more and higher-quality data, which we believe will have a very positive long-term impact on the community.
>
> > W2: Judge-based evaluation bias. The Stage-3 refinement risks evaluation circularity and bias toward the same models used for filtering.
>
> Thanks for the question. To mitigate this risk, we intentionally used different model families for the 'judge' backbone (GPT-OSS) and our primary generative model (Qwen-VL).
>
> As you requested, we randomly sampled 30K unfiltered data samples from Python and WebUI artifacts for cross-model adjudication. We evaluated them using three different judge models (Qwen2.5-VL-72B-Instruct, Gemini-2.5-pro, GPT-5), and computed the statistics and Krippendorff’s alpha across all judgments. The preliminary results are shown below:
>
> | | Qwen2.5-VL-72B (Mean) | Qwen2.5-VL-72B (Var) | Gemini2.5-Pro (Mean) | Gemini2.5-Pro (Var) | GPT-5 (Mean) | GPT-5 (Var) | Krippendorff's alpha |
> | :--- | :---: | :---: | :---: | :---: | :---: | :---: | :---: |
> | Python | 4.59 | 0.41 | 4.36 | 0.58 | 4.41 | 0.53 | 0.76 |
> | Artifacts | 4.28 | 0.47 | 4.05 | 0.61 | 4.16 | 0.52 | 0.71 |
>
> We observe that both categories yield a Krippendorff’s alpha above 0.70, indicating strong agreement in identifying data quality. This suggests that different backbones (open-source or commercial) can serve as a reliable filtering component in our pipeline. We will provide more comprehensive results in the final version.
>
> Furthermore, for future community use, reliability can be further enhanced by incorporating judgments from multiple models and applying techniques such as majority voting.
>
> > W3: Despite strong engineering effort, the paper reads largely as a dataset and data-pipeline contribution.
>
> Thank you for recognizing the strong engineering effort. Our core technical novelty is using a data-centric methodology to enable a unified visual-programmatic model. We posit that innovation is not confined to model architecture; the data strategy itself represents a fundamental technical contribution. Prior work was limited to specialized models for isolated tasks (e.g., chart-to-code or text-to-WebUI), largely because a diverse, large-scale corpus spanning these modalities was not available.
>
> Our data pipeline, which introduces (1) data types and (2) novel synthesis strategies (e.g., Bidirectional Translation) not seen in prior works, serves as the foundation that allows the model to harness cross-domain synergies. As analysis shows, our approach measurably boosts visual-coding performance, a key technical finding enabled by our approach that offers significant benefits to the community.
>
> Again, we thank you for your constructive review, which has helped us improve the quality of our paper. We will make our resources publicly available to help the community advance in this field. We hope our supplementary experiments and explanations can address your concerns.

---

> > ### Comment · Reviewer_NYgb · 2025-11-28
> >
> > Thank you for your response. My primary issues are resolved. I maintain my positive rating of this paper.

---

### Official Review · Reviewer_or79 · 2025-11-01

**Soundness:** 3
**Presentation:** 3
**Contribution:** 3
**Rating:** 6
**Confidence:** 4

**Summary:**

This paper presents JanusCoder, a suite of multimodal models aiming to unify code generation and visual reasoning through a “visual–programmatic interface.” The authors introduce a large-scale data synthesis toolkit and release JanusCode-800K, a multimodal corpus covering charts, WebUIs, visual artifacts, and animations. Two models—JanusCoder (text-centric) and JanusCoderV (vision-centric)—are trained on this corpus using Qwen and InternVL backbones. The authors also propose DTVBench, a benchmark for dynamic theorem visualization tasks. Experiments across seven benchmarks show that JanusCoder models outperform open-source baselines and sometimes rival GPT-4o. Ablation studies indicate that cross-domain synergies and reward-based data filtering contribute significantly to performance.

**Strengths:**

- Ambitious and timely goal: a unified model bridging code logic and visual semantics.

- The JanusCode-800K dataset appears large, diverse, and potentially impactful for future research.

- Strong empirical results on both text- and vision-centric benchmarks, including new ones created by the authors.

**Weaknesses:**

- Model novelty is limited. The architecture largely reuses Qwen and InternVL; the “unified interface” claim feels more conceptual than technical.

- Weak quantitative evidence for data quality improvements. The reward model and filtering pipeline are described but not systematically validated.

- Overextended scope. The paper attempts to be both a dataset, benchmark, and model paper, which dilutes its main scientific contribution.

- Lack of human evaluation for subjective visual tasks (animations, WebUIs).

**Questions:**

- What exactly differentiates the “unified visual-programmatic interface” from standard multimodal fine-tuning?
- Could you quantify the impact of reward modeling (e.g., pre- vs. post-filtering data quality metrics)?

---

> ### Author Response · Authors · 2025-11-18
> **Response to Reviewer or79**
>
> Thank you for your review. We appreciate your recognizing our work as an ambitious and timely goal, and our contributions to data and model performance. We address your questions as follows:
>
> > W1 & Q1: About “unified visual-programmatic interface”.
>
> Our contributions to building a unified visual-programmatic interface mainly feature:
> 1. Unified & Broad Task Spectrum: Unlike standard fine-tuning for isolated tasks (e.g., only chart2code, text-to-UI), our "unified interface" is a single model handling a heterogeneous spectrum of tasks (charts, WebUIs, dynamic animations). Achieving this generalization is *fundamentally a data challenge, not merely an architectural one*.
> 2. Novel Data: Our core technical contribution is the versatile data work that enables this unified capability. Innovation in foundation models is not limited to architecture; we build novel curation strategies (e.g., Bidirectional Translation) to build the largest, most diverse corpus of its kind, including previously scarce domains like large-scale animation data. Indeed, the data recipe itself is a core innovation, like the contributions of models like Phi-3 [1].
> 3. Cross-Domain Synergy: Further, our model leverages cross-domain synergies. Our analysis empirically validates that this unified training measurably improves performance on specialized visual-coding tasks, a benefit unattainable through isolated fine-tuning.
>
> > W2 & Q2:  About Quantitative evidence for data quality improvements. The reward model and filtering pipeline are described but not systematically validated. Could you quantify the impact of reward modeling?
>
> Thank you for this question. We provide a quantitative validation for the reward modeling pipeline in Section 6.1 Ablation Studies. The specific results are presented below. We compare our full models against versions trained without the reward modeling and filtering step (labeled 'w/o Rewarding').
>
> | | PandasPlotBench | PandasPlotBench | ArtifactsBench | LiveCodeBenchV6 |
> |---|---|---|---|---|
> | JanusCoder | 63 | 80 | 40.99 | 25.14 |
> ...
> | **w/o Rewarding** | 60 | 77 | 38.58 | 24.57 |
>
> | | ChartMimic | InteractScience | WebCode2M |
> |---|---|---|---|
> | JanusCoderV | 68.74 | 17.73 | 75.78 |
> ...
> | **w/o Rewarding** | 58.26 | 17.2 | 73.78 |
>
> As these results show, the effect of our reward model and filtering pipeline is clear for both text-centric and vision-centric tasks.
>
> > W3: The paper attempts to be both a dataset, benchmark, and model paper, which dilutes its main scientific contribution.
>
> We understand your concern regarding contribution dilution. Our work is tackling new, complex tasks and scenarios (unified visual-programmatic generation). As this is an emerging field, addressing it comprehensively requires these synergistic components to form long-term community benefit.
> Our data pipeline is a technical contribution that enables the development of our unified model. The benchmark is an auxiliary contribution developed to fill the critical evaluation gap for dynamic theorem visualization, which is necessary to fully assess our model's expanded scope.
>
> > W4: Lack of human evaluation for subjective visual tasks (animations, WebUIs).
>
> Following your suggestion, we conducted a small-scale additional human evaluation focused on subjective quality. This evaluation covered a total of 100 videos / demos, including video quality tests for Manim animations and visual quality tests for interactive WebUI demos (InteractScience). We used a [0-5] scale and invited 3 annotators with college backgrounds, providing them with evaluation guidelines and metrics adapted from the prompts used for our VLM Judge.
>
> | Model | Manim Video | Model | InteractScience WebUI |
> |---|---|---|---|
> | Qwen3-8b | 2.1 | Qwen-2.5-VL-7b | 0.92 |
> | Qwen3-14b | 2.64 | InternVL-3.5-4B | 1.67 |
> | Qwen-2.5-coder-7b | 2.48 | InternVL-3.5-8B | 1.57 |
> | Qwen-2.5-coder-32b | 2.82 | gpt4o | 2.78 |
> | JanusCoder-14B | 3.14 | JanusCoderV-7B | 3.19 |
>
> We will include more details of this human evaluation in the final version.
> Thank you again for your valuable feedback. We look forward to hearing from you again!
>
> ### Reference
>
> [1] Phi-3 Technical Report: A Highly Capable Language Model Locally on Your Phone

---

### Author Response · Authors · 2025-12-03
**General Response and Summary of Rebuttal Updates**

Dear Area Chair,

We sincerely appreciate the additional efforts invested in handling the submissions. To assist with your assessment, we summarize below the key facts regarding JanusCoder and the constructive author–reviewer discussions.
We confirm that we have engaged in the discussion strictly in accordance with the ICLR Code of Conduct.

During the discussion phase, we actively participated in conversations with all reviewers. We are pleased that all three reviewers maintained their positive ratings `6, 6, 8`.
Here, we summarize the discussion highlights and our key responses in the table below.

| Reviewer | Rating | Main Concerns / Request                                                    | Our Response                                                                                                             | Status                     |
|----------|---------------|----------------------------------------------------------------------------|--------------------------------------------------------------------------------------------------------------------------|----------------------------|
| `or79`     | 6             | Quantify reward modeling impact; lack of human eval for subjective tasks.  | Conducted new human evaluation (100 videos/demos); provided ablation data verifying reward model efficacy.               | Positive (No Reply)        |
| `NYgb`     | 6             | Data release plan; potential bias in judge-based evaluation (circularity). | Confirmed release plan; Conducted cross-model adjudication (GPT-5, Gemini-2.5-pro) showing high agreement ($\alpha > 0.7$). | Positive (Issues Resolved) |
| `uHBB`     | 8             | Scale of DTVBench; performance trade-offs on specific benchmarks.          | Clarified quality-focused curation; explained trade-off between functional correctness (TreeBLEU) and visual similarity. | Positive (No Reply)        |


Summary of Discussion Outcomes:
- Resolved Issues: Reviewer `NYgb` (6) responded to our rebuttal, confirming that their primary concerns were resolved after reviewing our new cross-model adjudication experiments (utilizing GPT-5 and Gemini-2.5-pro to validate judge reliability).
- New Evidence Provided: For Reviewers `or79` and `uHBB`, we provided the requested new human evaluation on subjective tasks and detailed ablation studies quantifying the impact of reward modeling. Given their initial positive ratings (6 & 8), we believe these additional results can effectively solidify the paper's contribution.

Once again, we thank the ACs for your time and attention. We hope this summary provides a clear picture of the discussion phase.

Best,

JanusCoder Authors.

---

### Meta-Review · Area_Chair_9Tfg · 2026-01-14

**Summary:**

This paper introduces JanusCoder, a suite of multimodal models designed to unify code generation and visual reasoning through a visual–programmatic interface. In particular, the authors present a large-scale data synthesis toolkit with two complementary models, one text-centric and the other vision-centric. While the addressed topic is timely and potentially impactful, several concerns were initially raised by the reviewers, including the need for clearer clarification of the model’s technical novelty.

**Reviewer Concerns:**

The reviewers raised concerns about the paper’s limited methodological novelty, noting that the model architecture largely reuses existing frameworks and that the claimed “unified interface” is more conceptual than technically substantive. Additional concerns include insufficient validation of data quality improvements, potential bias and circularity in judge-based evaluations, unclear plans for public data release affecting reproducibility, and a lack of human evaluation for subjective visual tasks. Reviewers also questioned the statistical reliability and coverage of the proposed benchmark, as well as unexplained performance gaps on certain datasets and missing discussions on broader reasoning-oriented visual tasks.

**Reviewer Scores:**

The main concerns highlighted during the review process involved limited technical innovation and potential bias in judge-based evaluations. During the rebuttal phase, however, these primary issues were largely addressed through clarifications and additional explanations.

---

### Decision · Program_Chairs · 2026-01-26

Accept (Poster)